# G3BP1 controls the senescence-associated secretome and its impact on cancer progression

Amr Omer [1,2], Monica Cruz Barrera [3], Julian L. Moran [1,2], Xian J. Lian[1,2], Sergio Di Marco[1,2], Christian Beausejour[3] & Imed-Eddine Gallouzi [1,2 ✉]

Cellular senescence is a known driver of carcinogenesis and age-related diseases, yet senescence is required for various physiological processes. However, the mechanisms and factors that control the negative effects of senescence while retaining its benefits are still elusive. Here, we show that the rasGAP SH3-binding protein 1 (G3BP1) is required for the activation of the senescent-associated secretory phenotype (SASP). During senescence, G3BP1 achieves this effect by promoting the association of the cyclic GMP-AMP synthase (cGAS) with cytosolic chromatin fragments. In turn, G3BP1, through cGAS, activates the NF-κB and STAT3 pathways, promoting SASP expression and secretion. G3BP1 depletion or pharmacological inhibition impairs the cGAS-pathway preventing the expression of SASP factors without affecting cell commitment to senescence. These SASPless senescent cells impair senescence-mediated growth of cancer cells in vitro and tumor growth in vivo. Our data reveal that G3BP1 is required for SASP expression and that SASP secretion is a primary mediator of senescence-associated tumor growth.

[1] Department of Biochemistry, McGill University, Montreal, QC H3G1Y6, Canada. [2] Rosalind & Morris Goodman Cancer Research Center, McGill University, Montreal, QC H3A1A3, Canada. [3] Centre de Recherche CHU Ste-Justine, and Department of Pharmacology and Physiology, Université de Montréal, Montréal, QC H3T1C5, Canada. ✉email: imed.gallouzi@mcgill.ca

The Ras GTPase-activating protein-binding protein 1 (G3BP1), a key factor in the stress response and stress granule (SG) assembly, is associated with several processes including pro-survival response and cell fate[1–5]. In mice, G3BP1 expression is essential, as its complete knockout leads to neonatal death[6]. In fact, $G3bp1^{-/-}$ mouse embryonic fibroblasts (MEFs) show an increase in cell death by apoptosis and impaired proliferation[6]. Recently obtained viable $G3bp1^{-/-}$ mice, from an alternate background, exhibit a premature aging phenotype as well as symptoms of pathologies related to aging such as ataxia[7]. While these observations suggest that G3BP1 could play a role in aging and age-related phenotypes, this possibility has yet to be explored.

One of the main promoters of age-related disease, such as cancer, is cellular senescence, a process by which cells enter an irreversible cell cycle arrest in response to various stresses[8–11]. Generally, these cells undergo profound molecular and biological changes, namely decreased genomic stability and chromatin organization due to a loss in Lamin B1 expression[12,13], induction of key cyclin-dependent kinase inhibitors (CDKIs) p21$^{WAF1/Cip1}$ and p16$^{INK4a}$ which inhibit proliferative signaling by retinoblastoma (RB) protein[8,14], increased markers of DNA damage[15–17] and induction of the senescent-associated secretory phenotype (SASP)[18]. The SASP is a large group of secreted factors that include cytokines, chemokines, angiogenic factors, extracellular matrix-remodeling proteases, and growth factors (e.g. IL-6, IL-8, and TNFα)[18,19]. Despite the protective role that senescence plays in an organism, the accumulation of senescent cells during aging has been associated with many cancers by enhancing neoplastic cell proliferation and metastasis[18,20,21]. The most striking evidence supporting a link between senescence and cancer is the fact that removing senescent cells from mice decreases cancer occurrence throughout their lifespan[22,23]. Since the loss of G3BP1 is associated with age-related phenotypes, it is possible that G3BP1 modulates these effects by controlling cellular senescence and cancer growth.

In this study, we assessed the role of G3BP1 as a regulator of the deleterious effects of senescent cells. Using in vitro and in vivo models of senescence and tumor growth, we determined that although G3BP1 does not affect the senescence process, it does play a vital role in the ability of senescent cells to induce the SASP. Importantly, we demonstrate that the SASP, in a G3BP1-dependent manner, are primary promoters of paracrine-mediated effects such as cancer growth. We also delineated the mechanism by which G3BP1 achieves this effect and we provide a proof-of-principle that G3BP1 could be considered as a potential drug target to alter the behavior of senescent cell in order to combat age-related disease.

## Results

**Loss of G3BP1 does not prevent cell commitment to senescence.** In order to determine the role of G3BP1 during cellular senescence, we used small interfering RNAs (siRNAs) targeting *G3BP1* mRNA in WI-38 or IMR-90 primary human lung fibroblasts, both of which are well-established cell models for cellular senescence[24,25]. We tested multiple siRNAs to target exons 1, 4 and 7 of *G3BP1* (Supplementary Fig. 1a), and found that the most consistent depletion was achieved using siRNA targeting exon 4 and exon 7 (referred hereafter as siG3BP1 #1, or simply siG3BP1, and siG3BP1 #2, respectively) which were utilized for the remainder of our study. We induced senescence by exposing WI-38 or IMR-90 expressing G3BP1 or not to ionizing radiation (10 Gy), or through lentiviral-mediated overexpression of HRAS$^{G12V}$ in WI-38 cells, and assessed primary markers of cellular senescence such as senescence-associated β-galactosidase activity (SA-

β-gal)[26], the loss of Lamin B1 (ref. [13]), and the formation of senescence-associated heterochromatin fragments (SAHF)[27]. We observed that depleting G3BP1 from WI-38 or IMR-90 did not affect their ability to commit to the senescence phenotype as demonstrated by an increase in SA-β-gal and SAHF formation 8-day post-ionizing radiation (+IR) and after HRAS$^{G12V}$ over-expression, as well as a decrease in Lamin B1 (*LMNB1*) mRNA steady-state levels (Fig. 1a–e and Supplementary Figs. 1b–d, 2a–c). These results were subsequently confirmed by examining early markers of senescence such as the accumulation of γH2AX foci in the nucleus[28] and the proliferative capacity of a cell using Ki-67 staining[29]. Indeed, the absence of G3BP1 promoted the formation of γH2AX foci in the nucleus and triggered a significant decrease in Ki-67 staining (Fig. 1e and Supplementary Fig. 1e–f). In addition, the expression of bona fide markers of senescence[30] such as p21$^{WAF1/Cip1}$ or p16$^{INK4a}$ did not change in the absence of G3BP1 when compared to siCTL-treated cells, while G3BP1 depletion had differential effects on the phosphorylation status of RB and p53 which was dependent on cell-type or the way by which we induced senescence (Fig. 1f, Supplementary Figs. 3, 4). Together these data clearly show that the depletion of G3BP1 from primary fibroblasts does not affect their ability to commit to senescence. Of note, since irradiation did not induce the formation of stress granules in WI-38 fibroblast (Supplementary Fig. 5), we also concluded that any effect of G3BP1 on the behavior of senescent fibroblasts, does not involve its well-known function as a promoter of stress granule assembly[3,31,32].

**Depletion of G3BP1 severely impairs the inflammatory SASP.** Next, we performed RNA sequencing analysis on total RNA from IR-treated WI-38 cells to determine the pathways which may be affected by the loss of G3BP1 in these senescent cells. Principal component analysis (PCA) of the top 5,000 most varying genes showed that depletion of G3BP1 substantially altered the levels of RNA in senescent cells (Fig. 2a). Moreover, as seen by hierarchical clustering of the top 500 most differentially expressed genes, this alteration was mainly due to a decrease in the levels of a majority of RNAs (Fig. 2b and Supplementary Data 1). We further analyzed this data using differential expression analysis and Ingenuity Pathway Analysis (IPA) in order to integrate our RNA sequencing data to known pathways. Assessing senescence related pathways, we found that the loss of G3BP1 drastically downregulated cytokine and immune response pathways when compared to cell cycle signaling (Fig. 2c–e). The most downregulated pathways were those involved in the promotion of SASP expression such as NF-κB, IL-8 (also known as CXCL8) and IL-6 signaling[19] (Fig. 2e–f). We validated targets of these pathways by qPCR and found that the loss of G3BP1 significantly decreased the levels of *TNFA*, *IL6* and *CXCL8* mRNAs in WI-38 and IMR-90 senescent cells (Fig. 3a, Supplementary Figs. 6a, 7a). We then assessed total levels of various SASP factors in conditioned media obtained from senescent WI-38 cells, expressing or not G3BP1, using multiplex arrays. Similar to our RNA sequencing analyses, G3BP1-depleted senescent cells had a reduction in a majority of SASP factors, notably many matrix metalloproteases and promoters of inflammation (Fig. 3b). We further examined NF-κB and STAT3 activation and signaling required for the promotion of the SASP[18] using western blot and immunofluorescence. G3BP1 depletion significantly reduced phosphorylation of IκBα and the nuclear localization of p65. STAT3 phosphorylation was also significantly reduced with the loss of G3BP1 (Fig. 3c–d, Supplementary Figs. 6b–c, 7b). Together, these observations demonstrate that the loss of G3BP1 downregulates NF-κB and STAT3 signaling and generates a unique class of senescent cells unable to produce inflammatory SASPs that we dub SASPless.

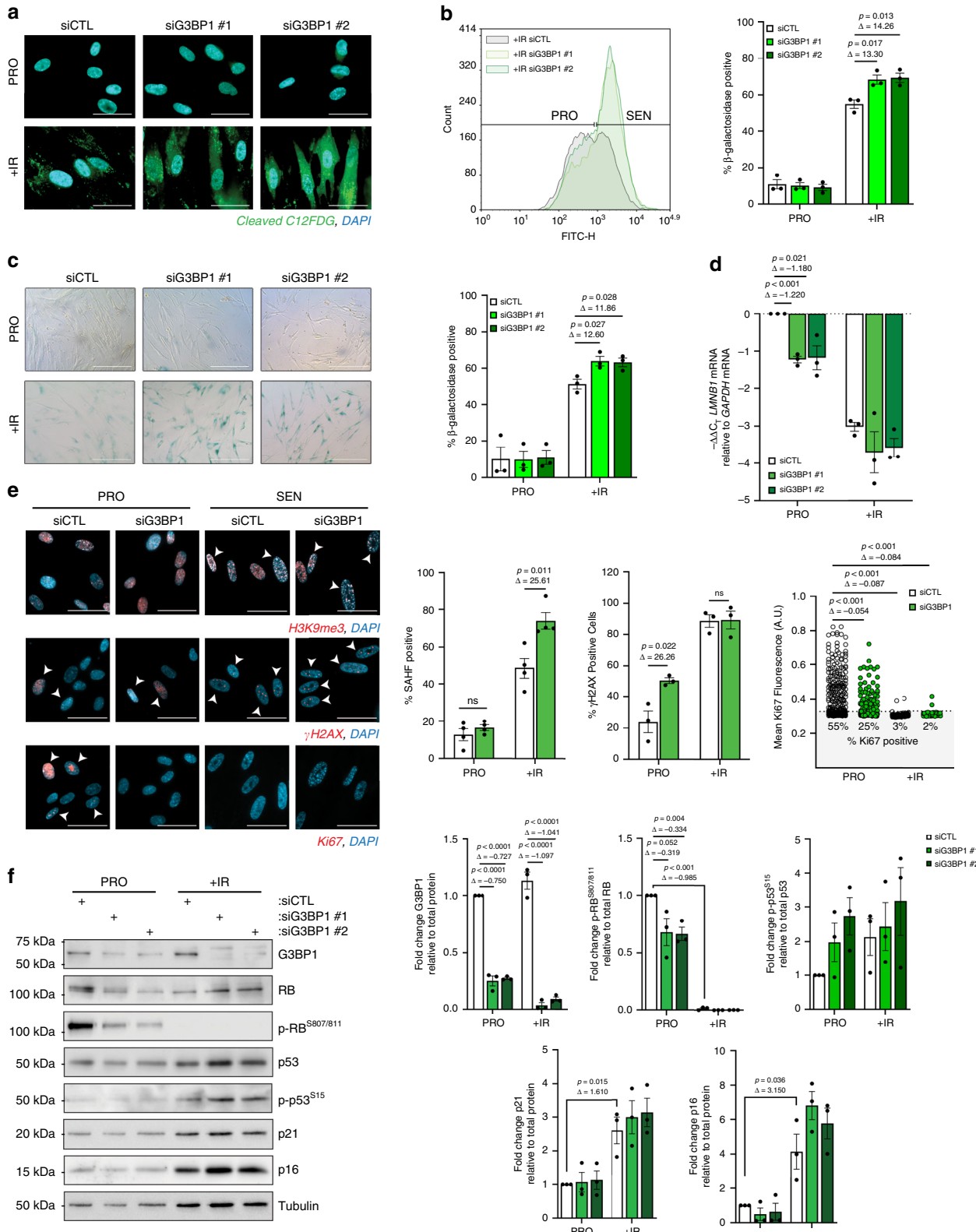

**G3BP1 induces the SASP through activation of the cGAS pathway.** The regulation and induction of the SASP is still unclear; however, recent observations demonstrated that the type I IFN response, activated by the cGAS-STING pathway, is required for the secondary activation of the SASP[33,34]. Since activation of cGAS requires G3BP1 (ref. [35]) we examined if the effect of G3BP1 on the production of SASPs during senescence

requires activation of cGAS and, consequently, the type I IFN response. We observed that, in both IMR-90 and WI-38 cell lines, although *IFNB1* mRNA levels were increased during senescence, these levels were ablated in G3BP1-depleted cells (Fig. 4a, Supplementary Figs. 8a, 9a). Transcriptional activation of IRF3, through its phosphorylation, is known to increase the expression of *IFNB1* mRNA[36]. Therefore, we examined if the activation of

**Fig. 1 G3BP1 depletion increases abundance of senescent cells and senescence-associated phenotypes.** WI-38 cells were treated with siRNA against G3BP1 (siG3BP1 #1 and #2) or scrambled control (siCTL) and assessed during proliferative stage (PRO) and 8-days post-ionizing radiation (+IR). **a** SA-β-gal was assessed using C12FDG. Cells were treated with DAPI to visualize nuclei. Scale bar, 50 μm. **b** (left) SA-β-gal was assessed using C12FDG via flow cytometry in +IR cells. (right) Graph representing % SA-β-gal positive cells using flow cytometer. The data are a mean of three independent experiments ± s.e.m. (two-tailed unpaired Student's *t* test). **c** (left) SA-β-gal was assessed using X-gal. Scale bar, 100 μm. (right) Graph representing % SA-β-gal positive cells from (**c**). The data are a mean of three independent experiments ± s.e.m. (two-tailed unpaired Student's *t* test, exact <0.001 *p*-values from left to right: 0.00016). **d** RNA was assayed by qPCR using primers against *LMNB1* mRNA. The data are a mean of three independent experiments ± s.e.m (two-tailed unpaired Student's *t* test) **e** WI-38 cells were analyzed by immunofluorescence against mH2A, γH2AX, and Ki67, as labeled, during PRO and +IR. DAPI staining was used to visualize nuclei. Arrows indicate cells positive for indicated marker. Scale bar, 50 μm. (right) Graphs representing % SAHF positive cells % γH2AX positive cells and mean Ki67 intensity. The data are a mean of four and three independent experiments ± s.e.m, for % SAHF positive cells, determined by formation of DAPI foci colocalized with H3K9me3 foci, % γH2AX positive cells, determined by the presence of >4 nuclear γH2AX foci >300 nm in diameter, respectively, and a distribution of measured Ki67 intensities of a minimum of 300 cells per condition, n = 376, 338, 300, 342 from left to right (two-tailed unpaired Student's *t* test, exact <0.001 *p*-values from left to right: 1.8E−13, 1.0E−15. 1.0E−15). **f** (left) WI-38 cell lysates were subjected to western blot analysis against indicated proteins. (right) Quantifications represent a mean of relative protein levels from three independent experiments ± s.e.m (two-tailed unpaired Student's *t* test, exact <0.001 *p*-values from left to right: 2.9E−6, 2.0E−7, 1.2E−6, 7E−8, 1.8E−8). Source Data for the graphs in (**b–f**) as well as blots in (**f**) are provided in the Source Data File.

IRF3 is affected by the depletion of G3BP1. Our data show that IRF3 phosphorylation was decreased in G3BP1-depleted senescent cells (Fig. 4b and Supplementary Fig. 9b). We next investigated whether cGAS activation during cellular senescence is responsible for the effect of G3BP1 on the SASP. Using immunofluorescence, we assessed whether the depletion of G3BP1 negatively affected the association of cGAS with cytosolic chromatin fragment (CCF). Depletion of G3BP1 dramatically decreased the colocalization of cGAS with CCFs when compared to control (Fig. 4c, d and Supplementary Fig. 8b, c), suggesting that loss of G3BP1 impairs the ability of cGAS to associate with CCFs. Despite the decrease in cGAS association, G3BP1-depleted cells demonstrated a ~10% increase in the number of cells positive for CCF (Fig. 4e and Supplementary Fig. 8d). Furthermore, senescent cells depleted of G3BP1 did not impact the number of CCFs formed per cell but significantly decreased the number of CCFs colocalized with cGAS per cell (Fig. 4f, g and Supplementary Fig. 8e, f). Therefore cGAS, in G3BP1-depleted senescent cells, is unable to associate with CCFs generated during cellular senescence, thereby impairing the type I IFN response.

Although the G3BP1-depleted cells had a decrease in the association of cGAS with CCFs during senescence, we sought to evaluate if G3BP1 acted upstream of cGAS through the simultaneous depletion of G3BP1 and cGAS. Indeed, loss of both G3BP1 and cGAS did not have any additive effects on cellular senescence, as assessed by SA-β-gal activity and loss of Lamin B1 (Supplementary Fig. 10a–d). We observed no additional effects of double depletion of G3BP1 and cGAS on steady-state levels of *IFNB1*, *TNFA*, *IL6*, and *CXCL8* mRNAs (Supplementary Fig. 10e). Furthermore, we evaluated whether the cGAS pathway could be recovered upon loss of G3BP1. To do this, we determined the impact of the supplementation of 2′3′ cGAMP, a product of cGAS, on the type I IFN response and the SASP. We found that supplementation of 2′3′ cGAMP upregulated the upstream phosphorylation of IRF3, IκBα and STAT3 relative to senescent cells depleted of G3BP1 (Fig. 4h). In addition, supplementation of 2′3′ cGAMP increased the steady-state levels of *IFNB1*, *TNFA*, *IL6*, and *CXCL8* mRNAs (Fig. 4i). Mechanistically, our data highlight that the effect of G3BP1 on the SASP is due to reduced cGAS activation resulting in the impairment of the type I IFN response.

To confirm the importance of G3BP1 function on SASP expression, we used epigallocatechin gallate (EGCG), an inhibitor of G3BP1 previously shown to impair G3BP1-cGAS interaction[35]. The addition of EGCG significantly reduced total IRF3 expression, and decreased phosphorylation of STAT3 and IκBα at 40μM in senescent WI-38 cells (Fig. 5a). We observed a decrease in the

mRNA levels of *IFNB1*, *TNFA*, *IL6*, and *CXCL8* mRNA, and a decrease in SASP secretion under these conditions (Fig. 5b, c). Importantly, we also showed that the effects of EGCG are G3BP1 dependent (Fig. 5d). Thus, G3BP1 inhibition by EGCG recapitulates the effects of G3BP1 depletion in senescent cells and represents a potential approach to inhibit expression of inflammatory SASP.

**G3BP1 depletion impairs SASP associated tumor growth.** One of the main impacts of the senescent cells includes the promotion of age-related diseases, such as tumor growth and metastasis[30]. Therefore, we first examined whether our SASPless senescent cells could promote tumor proliferation and migration in vitro. We used a transwell co-culture system in order to ensure that senescent cells and cancer cells were physically separated while sharing media and secreted soluble factors such as the SASP (Fig. 6a). As expected, WI-38 senescent cells secreting SASP significantly promoted the growth of A549 lung adenocarcinoma cancer cells[37]. However, G3BP1-depleted senescent cells did not affect cancer cell growth when compared to non-co-cultured A549 cells (Fig. 6b). The effect of senescent cells and SASPless senescent cells was also reproduced in immortalized skin fibroblasts derived from primary skin fibroblasts (CBSF-4T) co-cultured with WI-38 senescent cells expressing G3BP1 or not (Supplementary Fig. 11). We further examined the impact of G3BP1-depleted senescent cells on cancer cell migration in vitro. Transwell co-culture of WI-38 senescent cells depleted of G3BP1 significantly reduced cancer cell migration of A549 cells by ~15% when compared to control as demonstrated by wound healing assay after 24 h (Fig. 6c).

In addition, we sought to determine whether SASPless G3BP1-depleted cells are able to further impact surrounding cells in the microenvironment. Previous studies have shown that conditioned media from senescent cells was capable of promoting a senescence response in proliferative cells[34]. Therefore, using the same co-culture system described above, we co-cultured proliferative WI-38 cells with senescent cells depleted or not of G3BP1. G3BP1-depleted senescent cells were unable to promote senescence in proliferative WI-38 cells when compared to control siRNA treated senescent cells (Supplementary Fig. 12). These results suggest that G3BP1-depleted SASPless senescent cells lose their ability to promote paracrine-mediated effects in their surrounding microenvironment.

We next assessed the impact of G3BP1 depletion in senescent cells on the progression of tumors in vivo. Using immune-compromised NOD/SCID/IL2Rγ null (NSG) mice, we examined whether subcutaneous co-injection of cancer cells and senescent

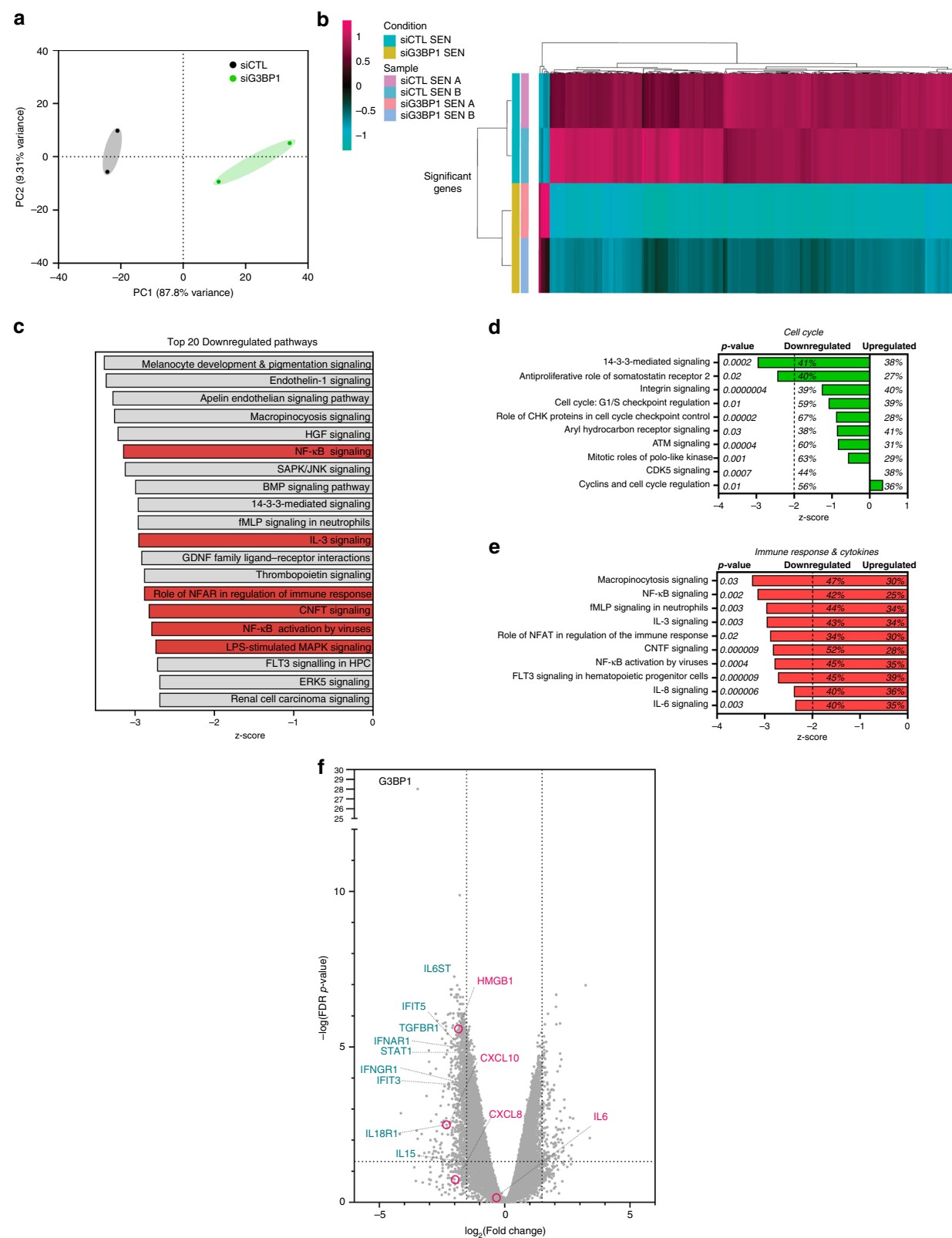

cells depleted or not of G3BP1 affected tumor growth and proliferation in vivo. We used A549 cancer cells, transduced with lentiviral vectors carrying the mPlum cDNA and senescent WI-38 cells with or without G3BP1. All WI-38 cells were stained with NIR790 cytoplasmic membrane dye, in order to allow for visualization of both the tumor and senescent cells using the Q-Lumi In Vivo imaging system (Fig. 7a). Furthermore, we

demonstrate that depletion of G3BP1 using siRNA is maintained for a minimum of 4-weeks (Supplementary Fig. 13a). Imaging of WI-38 senescent cells demonstrated that they were retained at the injection site within the growing tumor throughout the time period whereby we followed tumor growth (Fig. 7b and Supplementary Fig. 13b). As expected, the co-injection of A549 cells and senescent WI-38 cells increased the tumor volume when

**Fig. 2 RNA sequencing of senescent cells depleted of G3BP1 reveals decreased activation of cytokine and inflammatory signaling pathways. a** Total RNA from post-ionizing radiation (SEN) WI-38 cells treated with siRNA against G3BP1 (siG3BP1) or scrambled control (siCTL) were subjected to RNA sequencing. Data from the top 5000 varying genes from RNA sequencing were subjected to principle component analysis and scatterplot representing principle component 1 (PC1) and principle component 2 (PC2) are shown. **b** Data from RNA sequencing were subjected to hierarchical clustering analysis of top 500 most differentially expressed genes. **c** Data from RNA sequencing were subjected to Ingenuity Pathway Analysis and top 20 downregulated pathways (sorted by z-score with inflammatory signaling in red) are shown. **d, e** Data from RNA sequencing were subjected to Ingenuity Pathway Analysis and top 10 downregulated pathways specifically associated with cell cycle (**d**) and immune response (**e**) (sorted by z-score) are shown. *P*-values, % downregulated and % upregulated mRNAs in each pathway are shown. **f** Volcano plot of RNAs from RNA sequencing is shown (RNAs associated with inflammatory signaling — magenta, SASP associated RNAs — turquoise). Raw data for RNAseq are provided in Supplementary Data 1.

compared to mice injected with only A549 cells. However, G3BP1-depleted senescent WI-38 cells were unable to promote tumor growth when compared to control senescent WI-38 cells (Fig. 7c).

Using the integrated density determined by mPlum fluorescence we assessed whether the proliferation of tumor cells drove the increase in tumor volume. We observed a significant increase in the integrated density of tumors co-injected with control WI-38 senescent cells, but not G3BP1-depleted WI-38 senescent cells (Fig. 7d). Moreover, visual examination of the excised tumors as well as their mass show that the tumors co-injected with G3BP1-depleted senescent cells were smaller than their control counterparts (Fig. 7e). Therefore, these data demonstrate that SASPless senescent cells are unable to promote tumor growth in vitro and in vivo.

In summary, these findings implicate G3BP1 as a required factor during SASP induction in senescent cells. While the loss of G3BP1 does not affect cell commitment to senescence, it renders these cells SASPless. This outcome is due to the impairment of cGAS and the type I IFN pathway. In turn, these SASPless senescent cells do not promote cancer tumorigenesis and migration. Thus, G3BP1 plays a unique role in controlling the harmful behavior of senescent cells, and the targeting of G3BP1 represents a alternative strategy in combatting SASP-induced diseases.

## Discussion

The elimination of senescent cells or inhibition of senescence through the use of senolytics and senomorphics, as a strategy to combat cancer and age-related diseases, has proven to be a promising means to abolish the deleterious effects of the SASP on the promotion of tumor growth and metastasis. However, the absence of senescent cells in tissues comes at the high cost of disrupting critical physiological functions such as embryonic growth and patterning, glucose homeostasis, and wound healing[38–43]. Therefore, there is a pressing need to find ways to control the impact of senescent cells without their subsequent elimination. Here, we uncover G3BP1 as a key player in the production of SASP and delineate the molecular mechanism by which G3BP1 mediates this effect. While, as expected[35], G3BP1 is required for cGAS-mediated type I IFN response, our data show that the G3BP1-cGAS complex is essential for SASP production by senescent cells. We also demonstrate that the loss of G3BP1 during senescence generates a SASPless phenotype which impairs SASP-dependent tumor growth and metastasis in vitro and in vivo. Consequently, our data also show that the inflammatory SASP are primary drivers of deleterious effects of senescent cells, specifically regarding tumor growth. These findings, together with our data showing that G3BP1-depleted fibroblasts are still able to enter senescence, demonstrate that G3BP1 constitutes an alternative target to design a new generation of anti-cancer drugs able to prevent SASP production without affecting the senescence phenotype and its benefits.

Here we provide data showing that although the loss of G3BP1 enhances cell commitment to senescence, its loss had profound impact on the gene expression patterns of these cells. Indeed, the

depletion of G3BP1 decreased the expression levels of a majority of genes. Despite this, the normal expression patterns of bona fide promoters of cellular senescence, such as p21 and p16 were unaffected. Hence, it is not clear why G3BP1-depleted cells are more prone to commit to senescence than their control counterparts. One explanation for this phenotype could come from the fact that the knockdown of G3BP1 in proliferative cells triggers an immediate loss of Lamin B1. This observation raises the possibility that G3BP1 itself directly or indirectly regulates the expression of Lamin B1. Since G3BP1 has primarily been studied as an RNA-binding protein[4,5], it will be of high interest to investigate whether G3BP1 could mediate this effect on Lamin B1 expression by targeting *Lamin B1* mRNA post-transcriptionally. It is also possible that G3BP1 modulates other pro-senescence pathways. Indeed, the G3BP1-depleted cells exhibited early signs of increased DNA damage, as shown by the increase in p53 phosphorylation as well as increased presence of γH2AX foci. Hence, delineating the mechanisms by which G3BP1 modulates cell commitment to senescence is an important task that should be undertaken in the near future to help determine the impact of depleting G3BP1 on the behavior of senescent cells and their impact/interaction with their milieu.

In this study we delineate the mechanism by which G3BP1 modulates SASP activation in senescent cells. We demonstrate that loss or inhibition of G3BP1 significantly decreased the levels of *IFNB1, TNFA, IL6, CXCL8* mRNAs, and proteins. The downregulation of these factors is mediated by a reduction in cGAS association with CCFs found in G3BP1-depleted senescent cells. These observations are consistent with a previous report indicating that G3BP1 plays an essential role in the inflammatory response through its collaboration with pattern recognition receptors (PRRs). G3BP1 ensures the association of PRRs such as cGAS and RIG-1, to DNA and RNAs, respectively[35,44]. Additionally, accumulating evidence has demonstrated that cGAS is required for the induction of the type I IFN response and subsequent SASP activation[34,45]. Our observation showing that supplementation of 2′3′ cGAMP is sufficient to activate the type I IFN response and SASP in G3BP1-depleted cells, demonstrate that in senescent cells G3BP1 acts upstream of cGAS to promote CCF association and subsequent SASP production. However, while these observations provide a strong evidence that G3BP1 is required for cGAS-mediated production of SASP, they do not explain why G3BP1-depleted cells are still able to senesce. Our findings, together with the fact that cGAS is required for cell commitment to senescence, dependent on the cell type or means of senescence induction[34,46], indicate that cGAS can promote senescence independently of its effect on the SASP. Hence, it is likely that cGAS has a dualistic role in the induction of senescence and the SASP where under normal conditions cGAS can promote both senescence and SASP production. However, in the absence of G3BP1, cGAS loses its ability to promote SASP while preserving its pro-senescence activity. Exploring this possibility will help uncover mechanisms that could allow cells to enter senescence in a cGAS-dependent/SASP-independent manner and

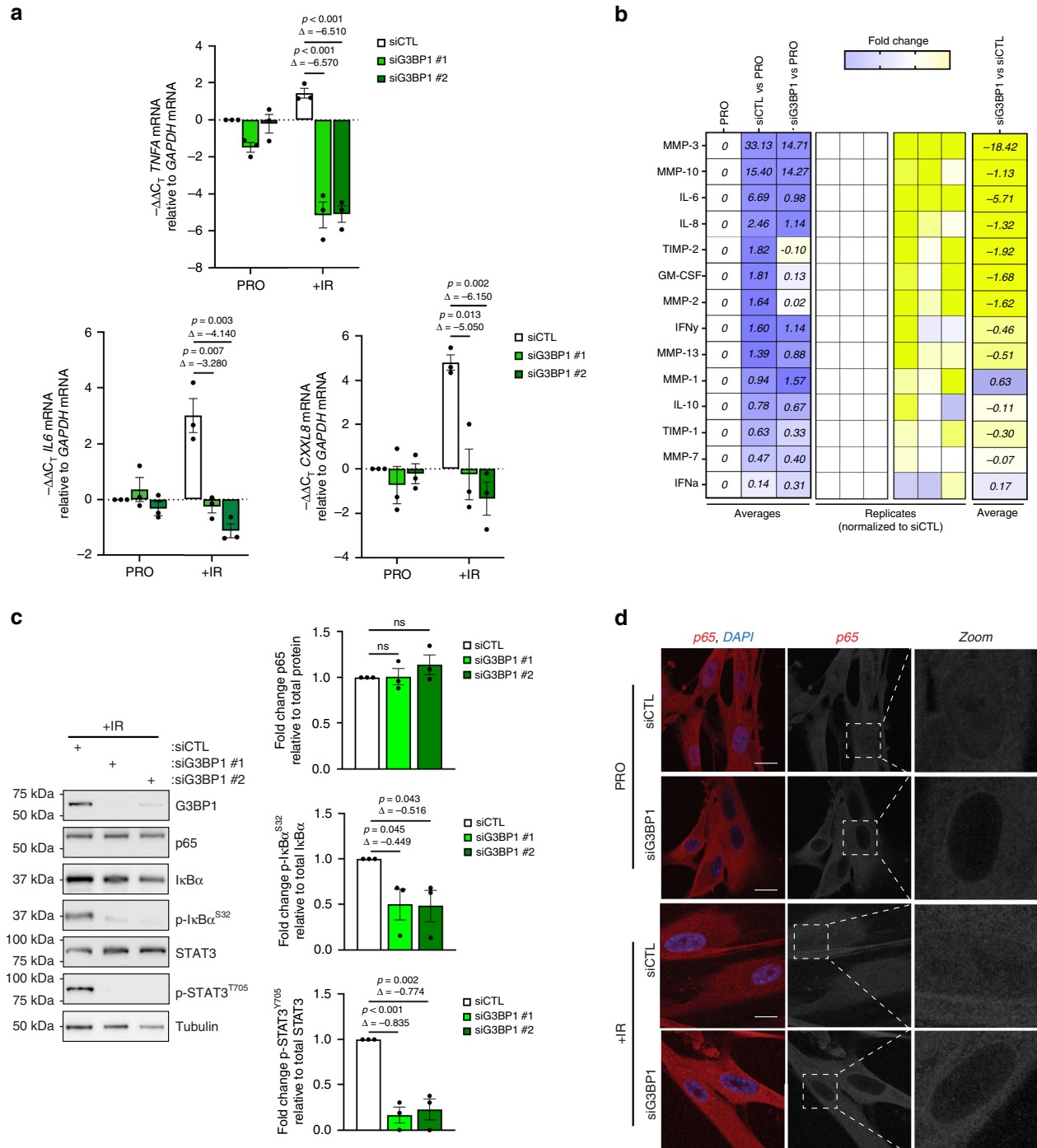

**Fig. 3 Depletion of G3BP1 impairs canonical SASP signaling through reduction of STAT3 and NF-κB signaling.** WI-38 cells were treated with siRNA against G3BP1 (siG3BP1 #1 and #2) or scrambled control (siCTL) and assessed during proliferative stage (PRO) and 8-day post-ionizing radiation (+IR). **a** RNA was extracted and assayed by RT-qPCR using primers against indicated mRNAs. The data are a mean of three independent experiments ± s.e.m (two-tailed unpaired Student's $t$ test, exact <0.001 $p$-values from left to right: 0.00095, 0.00022). **b** Conditioned media from PRO and from +IR WI-38 cells treated with siRNA against G3BP1 (siG3BP1) and scrambled control (siCTL) were analyzed by multiplex arrays. The heatmap indicates the fold change in comparison to the control PRO, SEN siCTL (siCTL), and SEN siG3BP1 (siG3BP1). Relative expression levels per replicate and average fold change differences are shown as indicated in heatmap. **c** (left) Cell lysates as described were subjected to western blot analysis against indicated proteins. (right) Quantifications represent a mean of relative protein levels from three independent experiments ± s.e.m (two-tailed unpaired Student's $t$ test, exact <0.001 $p$-values from left to right: 0.0007). **d** WI-38 cells as described were analyzed by immunofluorescence with antibody against p65 during PRO and SEN. DAPI staining was used to visualize nuclei. Scale bar, 20 μm. Source Data for panels (**a**) and (**c**) are provided in the Source Data File. Raw data for (**b**) is provided in Supplementary Data 2.

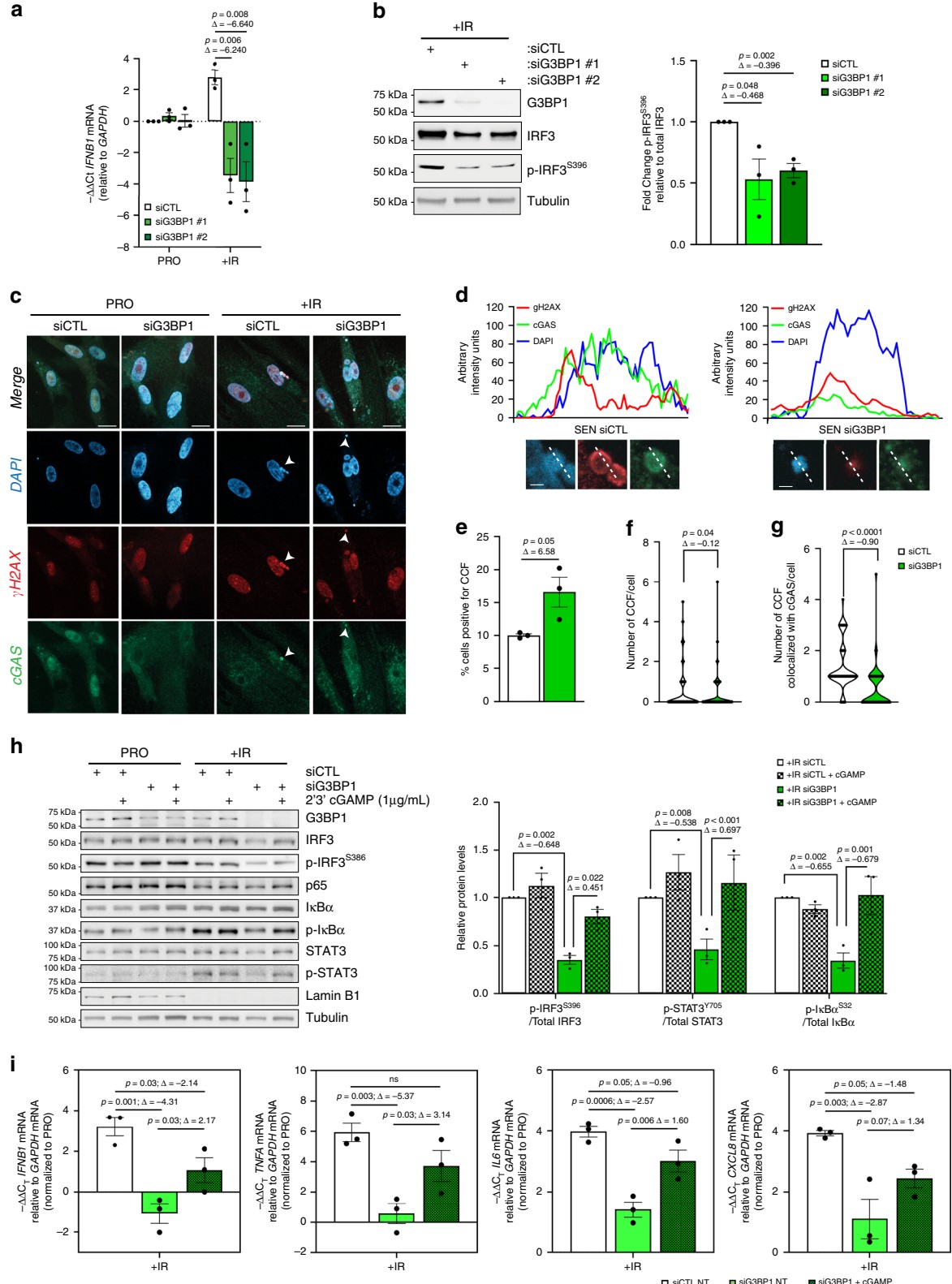

determine whether and how these cells preserve the benefits of senescence itself.

Without G3BP1, senescent cells are unable to promote their paracrine-mediated effects in vitro and unable to promote tumor cell proliferation in vitro and in vivo. While these data strongly support that the loss of G3BP1 prevents senescence-induced cancer, previous observations have shown that G3BP1 knockout mice

exhibit neurological defects, such as ataxia-related phenotypes, and premature aging[7]. These findings imply that the SASP is not the only way by which senescent cells exert their effects on the well-being of an organism as the loss of G3BP1 promotes age-related phenotypes in mice. The explicit removal of senescent cells or inhibition of senescence, while enticing, comes with negative repercussions in essential processes such as wound healing and

**Fig. 4 G3BP1 loss inhibits cGAS signaling required for SASP activation in senescent cells.** WI-38 cells were treated with siRNA against G3BP1 (siG3BP1 #1 and #2) or scrambled control (siCTL) and assessed during proliferative stage (PRO) and 8-day post-ionizing radiation (+IR) **a** RNA was extracted and assayed by RT-qPCR using primers against *IFNB1* mRNA. The data are representative of three independent experiments (two-tailed unpaired Student's *t* test). **b** (left) Lysates obtained from WI-38 cells +IR cell were subjected to western blot analysis against indicated proteins. (right) Quantifications represent a mean of relative protein levels from three independent experiments ± s.e.m (two-tailed unpaired Student's *t* test). **c** Immunofluorescence analysis against γH2AX and cGAS. DAPI staining was used to visualize nuclei. White arrows indicate CCF. Scale bar, 20 μm. **d** Graphs represent intensity profile for CCFs in (left) +IR siCTL and (right) +IR siG3BP1 treated WI-38 cells after immunofluorescence against γH2AX, cGAS, and DAPI. Plot is representative of foci shown in (**c**). Scale bar, 2 μm. **e–g** Graph of % SEN cells positive for CCF (**e**), number of CCFs present in SEN cells ± s.e.m (**f**), and number of CCFs present in SEN cells colocalized with cGAS (**g**). The data are representative of three independent experiments (two-tailed unpaired S tudent's *t* test, exact <0.001 *p*-values from left to right: 4.2E−7). **h** (left) Lysates obtained from WI-38 PRO and +IR cell treated with siRNA against G3BP1 (siG3BP1) or scrambled control (siCTL) were treated with 2'3' cGAMP (1 μg/mL) for 24 h and subjected to western blot analysis against indicated proteins. (right) Quantification of (left). Quantifications represent a mean of relative protein levels of +IR-treated cells from three independent experiments ± s.e.m (two-way ANOVA, Fisher's LSD, exact <0.001 *p*-values from left to right: 0.0009). **i** RNA prepared from PRO and +IR WI-38 cells treated with siRNA against G3BP1 or scrambled control (siCTL) were assayed by RT-qPCR using primers against the indicated mRNAs. The data are relative to PRO cells and are a mean of three independent experiments ± s.e.m (one-way ANOVA, Fisher's LSD). Source Data for panels (**a–b**) and (**d–i**) are provided in the Source Data File.

---

glucose homeostasis[38–40]. Similarly to G3BP1, other RNA-Binding proteins, such as PTBP1, have been shown to have the capacity to regulate the SASP without inhibiting senescence[47]. Unlike G3BP1, these factors were found to regulate SASP expression through alternative splicing. Whether or not the depletion of these specific proteins exerts a beneficial effect independent of SASP expression is still under debate. However, the benefit of G3BP1 depletion or depletion of factors required for the alternative splicing of the SASP on cancer progression is evident.

Our work goes further than the work of Georgilis et al., outlined above[47] by providing a potential therapeutic avenue to prevent SASP production without inhibiting cell commitment to senescence. We demonstrate that inhibition of G3BP1-cGAS interaction using the natural compound EGCG[35] is capable of impairing the SASP. Previous work has shown that EGCG is a potent anti-cancer compound[48–50]. Mechanistically, the effect of EGCG on cancer and senescent cells appears to differ, but treatment with EGCG impairs tumorigenesis independent of cell type. These observations suggest that EGCG represents a potent therapeutic with specific implications on G3BP1 in senescent cells impairing their ability to express the SASP. However, the fact that EGCG acts on various pathways, such as Wnt and AKT signaling[48–50], raises questions regarding its specificity and its use in a clinical setting. Therefore, further investigations are needed to identify specific compounds able to target G3BP1 in senescent cells and their implications on cancer progression, without the explicit removal or inhibition of senescence. If successful, such an endeavor will provide new generation of drugs for the treatment of cancer and age-related disease as alternatives to senolytics and senomorphics and their potential negative effects[51].

Overall, we demonstrate the requirement of G3BP1 in the induction of the SASP, but not the induction of senescence. Therefore, the ability of these cells to promote tumor growth is severely diminished due to the loss of SASP expression. How these cells interact with their environment, their ability to be eliminated by the immune system, or their direct impact on aging are still unclear. Moreover, the inhibition of the G3BP1-cGAS interaction using drugs such as EGCG, is a clear indication that G3BP1 could be a viable, and a more efficient target to prevent SASP secretion without affecting commitment of cells to senescence. Potentially targeting senescent cells using similar drugs could provide an alternative and less detrimental means to alleviate cancer and other age-related disease.

## Methods

**Cell culture**. WI-38 and IMR-90 cells were purchased from the European Collection of Authenticated Cell Cultures (ECACC Cat# 90020107) and American

Type Culture Collection (ATCC Cat# CCL-186), respectively. A549 and Immortalized Skin Fibroblasts (CBSF-4T) were kindly gifted to us from the laboratory of Dr. Christian Beausejour at the CHU Sainte-Justine Research Center. HEK 293 T cells were purchased from the American Type Culture Collection (ATCC Cat# CRL-3216). All cell lines were grown in a 5% $CO_2$ environment at 37 °C in Dulbecco's Modified Eagle's Medium (DMEM) (Invitrogen) supplemented with 10% fetal bovine serum (FBS) (Sigma) and 1% penicillin/streptomycin antibiotics (Sigma). WI-38 and IMR-90 cells were induced into senescence by exposure to ionizing radiation (10 Gy) or lentiviral transduction (empty vector, EV, or $HRAS^{G12V}$) when cells were ~70% confluent.

**Lentiviral vector production and transduction**. HEK 293T cells were transfected with psPAX2, pMD2.G plasmids and the lentiviral vector (pLenti CMV RasV12 Neo or pLenti CMV Blast empty) using Fugene 6. The supernatant containing lentiviral particles was harvested at 48 and 72 h. WI-38 cells were transduced with the lentiviruses through their direct addition to the culture medium and combination with Polybrene at 8 μg/μL. pLenti CMV RasV12 Neo (w108-1) was a gift from Eric Campeau (Addgene plasmid #22259) and pLenti CMV Blast empty (w263-1) was a gift from Eric Campeau & Paul Kaufman (Addgene plasmid #17486).

**In vivo tumor growth assay**. A549 lung carcinoma cells were transduced with lentiviral vectors carrying the mPlum cDNA along with a puromycin selection cassette. mPlum was subcloned from pQC mPlum XI generated by Connie Cepko (Addgene Cat# 37355). NOD/SCID/IL2Rγ null (NSG) mice were obtained from the Jackson Laboratory (Bar Harbor, ME) and housed in the animal care facility at the CHU Sainte-Justine Research Centre under pathogen-free conditions in sterile ventilated racks between 20 and 24 degrees Celsius at 40–70% humidity (typically 50%) using a 12 h dark/light cycle. All in vivo manipulations were previously approved by the CHU Sainte-Justine Research Centre institutional committee for good laboratory practices for animal research (Protocol #579). For tumor formation, $2.5 \times 10^5$ A549 mPlum cells were injected subcutaneously either alone or in combination with $5 \times 10^5$ WI-38 cells induced into senescence by exposure to ionizing radiation (10 Gy) and transfected or not with siRNAs against G3BP1 or control siRNA from Dharmacon. Injections were done in 50 μL of RPMI (Wisent Cat# 350-000-EL) to form a lump under the skin of anesthetized mice previously shaved and wet with alcohol. Two subcutaneous injections per mouse (on the left and right side of the back) were performed. Tracking of senescent WI-38 cells was possible through staining of the cells with CellBrite™ NIR790 Cytoplasmic Membrane Dye 786/820 nm (Biotium Cat# 30079) prior to injection. Monitoring of WI-38 and mPlum-expressing tumors was done twice weekly using the Q-Lumi In Vivo imaging system (MediLumine, Montreal) using the following filters, respectively (Ex.769-41 nm/Em.832-37 nm and Ex.562-40 nm/Em.641-75 nm). The fluorescence signal was standardized internally for each picture and normalized using ImageJ macros for picture processing. Analysis of tumor signal was also measured semi-manually using ImageJ macros and expressed in fluorescence integrated density.

**Transfection**. WI-38 and IMR-90 cells were transfected with siRNA using jet-PRIME (Polyplus transfection reagent) following the manufacturer's protocol when the cells reached 70-80% confluency. Cells were either collected (PRO) or induced into senescence 24 h after transfection and collected as described in each figure legend.

**siRNA**. siRNA specific for G3BP1 (exon 1) (5'-UGCCGUUCACUUUCUAUGC AA-3'; siRNA ID #n332329) was obtained from Ambion. siRNA specific for

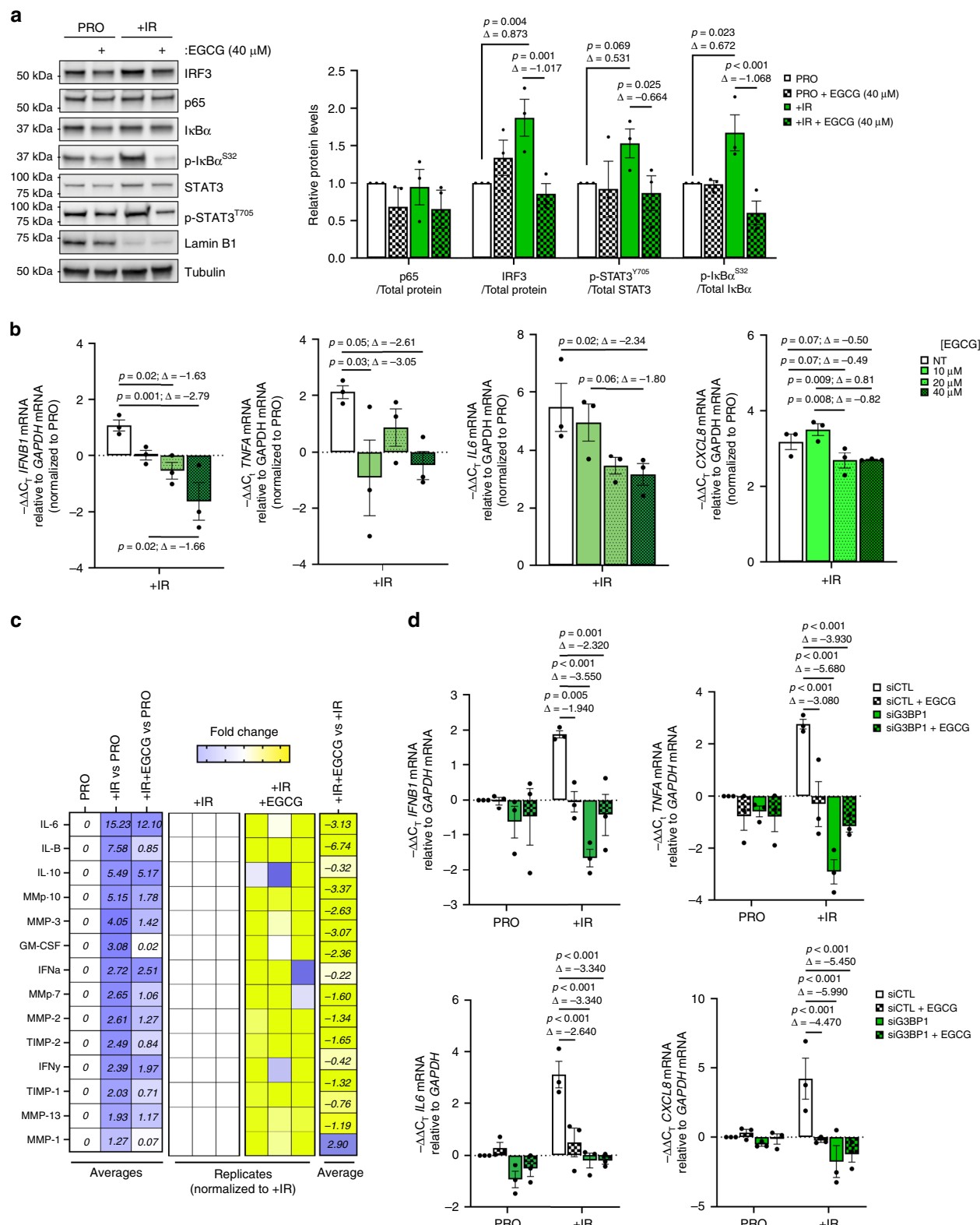

G3BP1 (exon 4 or siG3BP1 or siG3BP1 #1) (5′-UCAACAUGGCGAAUCUUG GTG-3′; siRNA ID #126650) was obtained from Ambion. siRNA specific for G3BP1 (exon 7 or siG3BP1 #2) (5′-UACUGGCUCAGGUUUUCCTC-3′; siRNA ID #126651) was obtained from Ambion. Control siRNA was obtained from Dharmacon (5′-AAGCCAAUUCAUCAGCAAUGG-3′).

**β-galactosidase staining**. For immunofluorescence, cells were stained for β-galactosidase activity using 5-Dodecanoylaminofluorescein di-β-D-galactopyranoside (C$_{12}$FDG) (ThermoFisher, Cat# D2893). Cells were treated for

1 h with bafilomycin A1 (Sigma-Aldrich, Cat# B1793-2UG), which disrupts lysosome formation leading to diffusion of β-galactosidase into the cytosol, and then with C$_{12}$FDG for 2 h. Before fixation, cells were stained with DAPI and immunofluorescence was performed using the Zeiss Axio Observer.Z1. For light microscopy, the Senescence Cells Histochemical Staining Kit (Sigma Catalog # CS0030) was used to detect senescent cells following the manufacturer's protocol.

**Quantification of β-galactosidase staining**. WI-38 or IMR-90 cells were seeded at 70–80% confluence in 96-well culture plates, cultured and induced into

**Fig. 5 The G3BP1 inhibitor, EGCG, recapitulates SASP inhibition associated with G3BP1 loss. a** (left) Cell lysates from proliferative and 8-day post-ionizing radiation (+IR) WI-38 cells treated or not with EGCG (40 μM) were subjected to western blot analysis against indicated proteins. (right) Quantifications represent a mean of relative protein levels from three independent experiments ± s.e.m (two-way ANOVA, Fisher's LSD). **b** RNA was extracted from cells during proliferative stage (PRO) and 8-day post-ionizing radiation (+IR) from WI-38 cells treated with or without (NT) increasing concentrations of EGCG (10, 20, and 40 μM) and assayed by RT-qPCR using primers against the indicated mRNAs. The data are relative to PRO cells and are a mean of three independent experiments ± s.e.m (one-way ANOVA, Fisher's LSD). **c** Conditioned media from proliferative (PRO) cells and 8-day post-ionizing radiation (SEN) WI-38 cells treated with or without 40 μM EGCG. The heatmap indicates the fold change in comparison to the control PRO, SEN, and SEN treated with 40 μM EGCG. Relative expression levels per replicate and average fold change differences are shown as indicated in heatmap. The data are representative of three independent experiments. **d** RNA was extracted from cells treated with siRNA targeting G3BP1 (siG3BP1) or scrambled control (siCTL) during proliferative stage (PRO) and 8-day post-ionizing radiation (+IR) from WI-38 cells in the presence or absence (NT) of EGCG (40μM). RT-qPCR was then performed using primers against indicated mRNAs. The data are a mean of three independent experiments ± s.e.m (two-way ANOVA, Fisher's LSD, exact <0.001 p-values from left to right: (top) 0.0001, 0.0001, 0.0001, 0.0011, (bottom) 0.0001, 0.0001, 0.0001, 0.0004, 0.0001, 0.0001). Source Data for panels (**a**), (**b**), and (**d**) are provided in the Source Data File. Raw data for (**c**) is provided in Supplementary Data 3.

senescence as described above. Proliferative cells and senescent cells (8-days post-irradiation or post-HRAS$^{G12V}$ lentiviral transduction) were stained for β-galactosidase activity as described above. Images were then immediately captured using the IN Cell Analyzer 2000. The acquired images were analyzed using the Multi Target Analysis Module that allows the creation of various decision trees and the application of appropriate classification filters to different image stacks. All samples were analyzed in triplicate with 9 fields per well.

**Flow Cytometry**. Senescent WI-38 or IMR-90 cells were seeded at 70–80% confluence in 6 cm culture plates, cultured and induced into senescence as described above. Eight days post-irradiation or post-HRAS$^{G12V}$ lentiviral transduction, lysosomal alkalinisation was induced by pre-treating cells with 100 nM bafilomycin A1 for 1 h in fresh cell culture medium at 37 °C and 5% CO2. C12FDG (33 μM) solution was then added to the cell culture medium for 2 h. The cells were harvested by trypsinization and resuspended 1X PBS. The cells were analyzed by flow cytometry within 1 h. To estimate relative SA-β-Gal activity, a two-parameter display of FSC vs. SSC was set up excluding subcellular debris. Single cells were selected using a two-parameter display of FSC-H vs. FSC-A. Non-labeled samples were used to determine auto-fluorescence. The gating strategy is shown in Supplementary Fig. 14.

**Immunofluorescence**. WI-38 and IMR-90 cells grown on coverslips were fixed and immunofluorescence experiments were performed using antibodies against H3K9me3 (Abcam Cat# ab8898, 1/500), phospho-histone H2A.X (Ser139/Tyr142) (Cell Signaling Technology Cat# 5438), Ki-67 (Abcam Cat# ab16667, 1/250), p65 (Abcam Cat# ab16502, 1/250) and cGAS (Thermo Fisher Scientific Cat# 14-5158-82, 1/200). Images were obtained using a Laser Scanning Microscope (LSM) 800 Confocal Microscope (Advanced Bio-imaging Facility, McGill University) or the Zeiss AxioObserver Z.1 inverted microscope.

**Real-time PCR**. TRIzol was used to extract RNA from cells. Following RNA extraction, RNA quantity and quality was determined by spectrophotometric analysis using a Thermofisher NanoDrop™ reader (ND-1000). A reverse transcription reaction was performed using 1ug of total RNA and iScript cDNA Synthesis Kit (Bio-rad Cat# 1708891). A negative control lacking the reverse transcriptase was also performed. Generated cDNA was then analyzed by qPCR using SsoFast EvaGreen Supermix (Bio-rad Cat# 172-5203) and quantified using a Corbett RG-6000. All levels were normalized to GAPDH and subsequently relativized to the non-treated control for all experimental replicates[52]. Primers used are listed as follows:

*LMNB1 Forward* (5′-AAGGCGAAGAAGAGAGGTTGAAG-3′)
*LMNB1 Reverse* (5′-GCGGAATGAGAGATGCTAACACT-3′)
*TNFA Forward* (5′- CTGCCCCAATCCCTTTATT-3′)
*TNFA Reverse* (5′-CCCAATTCTCTTTTTGAGCC-3′)
*IL6 Forward* (5′-CAGGAGCCCAGCTATGAACT-3′)
*IL6 Reverse* (5′-GAAGGCAGCAGGCAACAC-3′)
*CXCL8 Forward* (5′-GTGCAGTTTTGCCAAGGAGT-3′)
*CXCL8 Reverse* (5′-CTCTGCACCCAGTTTTCCTT-3′)
*IFNB Forward* (5′-AGGACAGGATGAACTTTGAC-3′)
*IFNB Reverse* (5′-TGATAGACATTAGCCAGGAG-3′)
*GAPDH Forward* (5′-TTGATTTTGGAGGGATCTCG-3′)
*GAPDH Reverse* (5′-TCACCAGGGCTGCTTTTAAC-3′).

**Western blot analysis**. Western blot was performed using total protein extracts and probing with antibodies against G3BP1 (Homemade, 1/5000), Tubulin (Developmental Studies Hybridoma Bank, 1/5000). Lamin B1 (Abcam Cat# ab16048, 1/10000), RB (Abcam Cat# ab181616, 1/1000), pRB (S807/811) (Cell Signaling Technology Cat# 8516, 1/1000), p53 (Millipore Cat# OP43-100UG, 1/500), pp53 (S15) (Cell Signaling Technology Cat# 9284, 1/1000), p21 (Abcam Cat# ab7960, 1/2000), p16 (Abcam Cat# ab108349, 1/1000), p65 (Abcam Cat# ab16502,

1/5000), STAT3 (Cell Signaling Technology Cat# 9139, 1/1000), pSTAT3 (Y705) (Cell Signaling Technology Cat# 9145, 1/1000), IκBα (Cell Signaling Technology Cat# 4814, 1/1000), pIκBα (S32) (Abcam Cat# ab92700, 1/1000), IRF3 (Cell Signaling Technology Cat# 11904, 1/1000), pIRF3 (S396) (Cell Signaling Technology Cat# 4947, 1/1000).

**Total RNA sequencing**. Total RNA from senescent WI-38 cells treated with siRNA targeting G3BP1 (siG3BP1) or scrambled control (siCTL) was isolated using TRIzol reagent (Invitrogen). RNA samples were assessed for quantity and quality using a NanoDrop UV spectrophotometer (Thermo Fisher Scientific Inc), and a Bioanalyser (Agilent Technology Inc). The 4 RNA-seq libraries were sequenced on the Illumina NextSeq 500 platform at the Institute for Research in Immunology and Cancer (IRIC) Genomics Core Facility, University of Montreal, to produce over 60 million, 100 nucleotide paired-end reads per sample. Sequences were trimmed for sequencing adapters and low quality 3′ bases using Trimmomatic version 0.35 and aligned to the reference human genome version GRCh38 (gene annotation from Gencode version 26, based on Ensembl 88) using STAR version 2.5.1b. Gene expressions were obtained both as readcount directly from STAR as well as computed using RSEM in order to obtain gene and transcript level expression, either in TPM or FPKM values, for these stranded RNA libraries. DESeq2 version 1.22.1 was then used to normalize gene readcounts. Sample clustering based on normalized log readcounts produces the following hierarchy of samples. (Data can be accessed in GEO database, GSE151745).

**Ingenuity pathway analysis**. RNA Sequencing Data were analyzed through the use of IPA (QIAGEN Inc., https://www.qiagenbioinformatics.com/products/ingenuity- pathway-analysis).

**Multiplex protein analyses**. Proinflammatory cytokine and chemokine protein levels in condition media were measured using Luminex xMAP technology by Eve Technologies Corp. (Calgary, Alberta, Canada). Human multiplex kits were from Millipore (Billerica, MA).

**Cyclic GMP-AMP supplementation**. Mammalian cyclic [G(2′,5′)pA(3′,5′)p] (cGAMP) (Invivogen, Cat# tlrl-nacga23) was dissolved in endotoxin-free water at a stock concentration of 1 mg/mL. For supplementation experiments, cGAMP was added directly to cell media to a concentration of 1ug/mL.

**Epigallocatechin gallate treatment**. Epigallocatechin gallate (EGCG) (Sigma, Cat# E4143) was dissolved in endotoxin-free water at a stock concentration of 10 mM. For G3BP1 inhibition experiments, EGCG was added directly to cell media at concentrations described in corresponding figure legends.

**Transwell cancer growth experiments**. WI-38 cells were growth to 70–80% confluency on transwell permeable supports (Costar, Cat# 3412), treated with control siRNA or siRNA against G3BP1 and irradiated. Eight days following irradiation, 50,000 A549 or CBSF-4T cell lines were plated on 6-well dishes and supports containing no cells, or either control siRNA or siRNA against G3BP1 treated cells were added above the cells. Total cell number was counted every 24 h using a Beckman Z-series Coulter Counter after trypsinization.

**Transwell in vitro scratch experiments**. Transwell plates were prepared as described above. Briefly, A549 cells were plated on 6-well dishes and grown to confluency. Scratches were made using a p200 pipette tip. Debris was removed by washing cells in cell media. Media was then replaced, and initial scratches were imaged using a phase-contrast microscope. Transwell plates were added to culture plates and after 24 h scratches were imaged using a phase-contrast microscope.

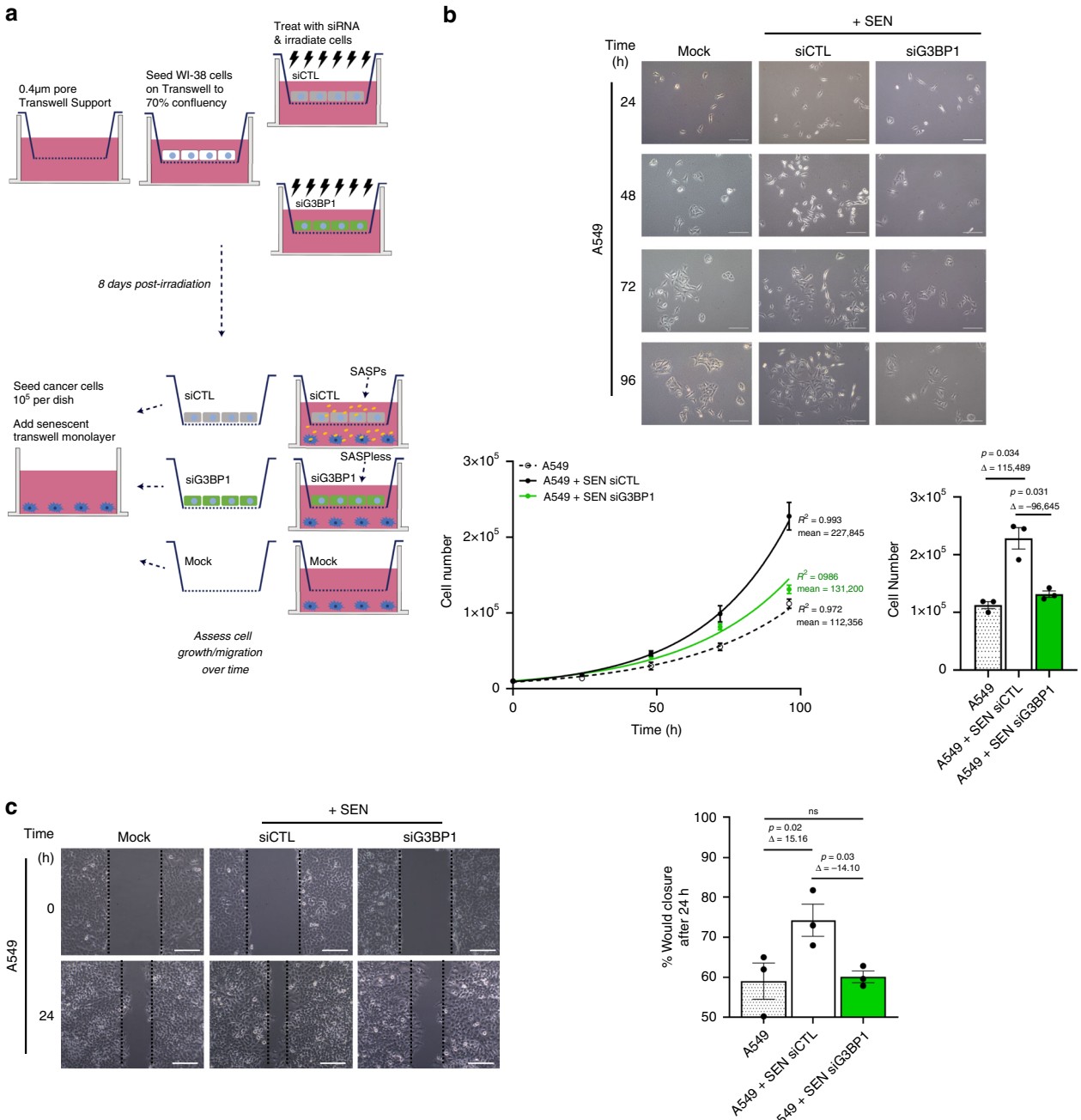

**Fig. 6 SASPless G3BP1-depleted senescent cells do not promote senescence-associated tumor growth in vitro. a** Schematic of transwell support co-culture experiments to assess impact of SASPs from 8-day post-ionizing radiation (SEN) WI-38 cells treated with siRNA against G3BP1 (siG3BP1), scrambled control (siCTL) on A549 cell growth. Mock cultures (no cells) was used as a control. **b** (top) Images of A549 cells without co-culture or with co-culture of SEN cells treated with siRNA against G3BP1 (siG3BP1) or scrambled control (siCTL). Scale bar, 200 μm. (bottom, left), Graph representing total number of A549 cells quantified over time without co-culture or with co-culture of SEN cells treated with siRNA against G3BP1 (siG3BP1) or scrambled control (siCTL). The data are a mean of three independent experiments ± s.e.m (nonlinear regression, R-squared and mean). (bottom right) Graph representing total number of A549 cells in (**b**, bottom, left) after 96 h. The data are a mean of three independent experiments ± s.e.m (one-way ANOVA, Fisher's LSD). **c** (left) Images of A549 cells immediately after scratch (0 h) and 24 h after scratch (24 h) without co-culture or with co-culture of SEN cells treated with siRNA against G3BP1 (siG3BP1) or scrambled control (siCTL). Scale bar, 100 μm. (right) Graph representing % wound closure without co-culture or with co-culture of SEN cells treated with siRNA against G3BP1 (siG3BP1) or scrambled control (siCTL) after 24 h. The data are a mean of three independent experiments ± s.e.m (one-way ANOVA, Fisher's LSD). Source Data for the graphs in panels (**b**) and (**c**) are provided in the Source Data File.

One hundred readings were taken and percentage wound closure was measured using Image J.

**Statistics and reproducibility**. GraphPad Prism 7.0 was used for statistical analyses. Two-tailed Student's *t* tests were used to determine statistically significant differences between two groups. One-way ANOVA or two-way ANOVA (Fisher's LSD test) was used for multiple comparisons. Nonlinear regression was performed using least square regression and considering the mean of each point using $1/Y^2$ weighting. R-squared values are displayed. *P*-values (*p*) and effect sizes (Δ) are displayed on all bar graphs and curves. For ANOVAs, *F* values and degrees of freedom are provided in the Source Data file. For *t* tests, *t* values and degrees of freedom are provided in the Source Data file. All images from Figs. 1a, 3d, 4c, 4d and Supplementary Figs. 2a, 5, 6c, 8b, 8c are representative of independently reproduced experiments. Each experiment was reproduced three times.

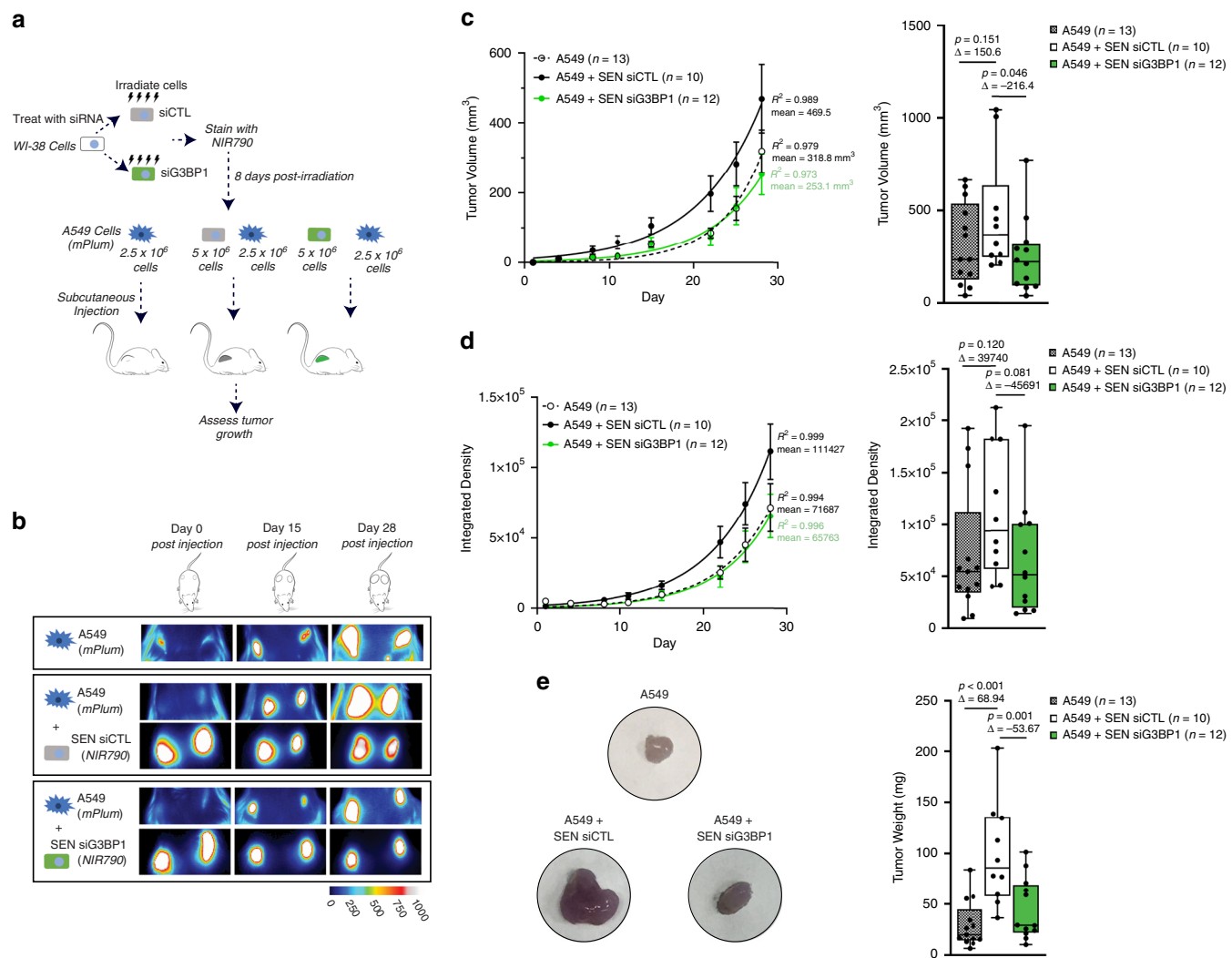

**Fig. 7 Co-injection of senescent cells depleted of G3BP1 with tumorigenic cells do not promote senescence-associated tumor growth in vivo.**
**a** Schematic representation of co-injection model in NSG mice. **b** Images of fluorescent signal from A549 injected alone (expressing *mPlum*) as well as co-injected A549 cells (expressing *mPlum*) and senescent WI-38 cells treated with siRNA targeting scrambled control (SEN siCTL, stained with NIR790) or A549 cells (expressing *mPlum*) and senescent WI-38 cells treated with siRNA targeting G3BP1 (SEN siG3BP1, stained with NIR790) were obtained using the Q-Lumi In Vivo imaging system. Images from day 0, day 15, and day 28 are shown. **c** (left) Graph representing tumor volume over time from independent tumors. The data are a mean of *n = 13 tumors* (A549 alone), *n = 10 tumors* (A549 co-injected with SEN siCTL), and *n = 12 tumors* (A549 co-injected with SEN siG3BP1) ± s.e.m (nonlinear regression, R-squared and mean). (right) Graph representing tumor volumes 28 days as in (left). (right) Box and whiskers plot representing tumor volume after 28 days as in (left) **d** (left) Graph representing integrated density over time from independent tumors. The data are a mean of *n = 13 tumors* (A549 alone), *n = 10 tumors* (A549 co-injected with SEN siCTL), and *n = 12 tumors* (A549 co-injected with SEN siG3BP1) ± s.e.m (nonlinear regression, R-squared and mean). (right) Box and whiskers plot representing integrated density after 28 days as in (left). **e** (left) Representative images of tumors at day 32. (right) Box and whiskers plot representing tumor weights of tumors from NSG mice at day 32. (**c, d, e,** right) The box and whiskers plot in these figures extends from the 25th to 75th percentile, the line within the box represents the median and the whiskers extend from min to max (one-way ANOVA, Fisher's LSD). Source Data for panels (**c**), (**d**), and (**e**) are provided in the Source Data File.

**Reporting summary**. Further information on research design is available in the Nature Research Reporting Summary linked to this article.

## Data availability

The data reported in this study in support of all the findings outlined are available from the corresponding author upon reasonable request. The raw multiplex analysis data for Figs. 3b and 5c are available in Supplementary Data 2 and 3, respectively. The raw RNASeq data are available in Supplementary Data 1 file and have been deposited into NCBI Gene Expression Omnibus (GEO) database under accession number GSE151745. For RNA Sequencing Analysis, sequences were aligned to the reference human genome version GRCh38 (gene annotation from Gencode version 26, based on Ensembl 88) using STAR version 2.5.1b. Source data are provided with this paper.

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

## Acknowledgements

The authors would like to acknowledge Esther Nam, Romane Monnet, Amanda Centorame and Farah Ben Brahim for their technical support throughout this work. We would like to acknowledge Ahmed Omer for his help in creating original digital images, specifically in Fig. 7 a, b. This work is funded by a Canadian Institute of Health Research (CIHR) project grant (PJT-159615) to I.E.G. and (MOP-341566) to C.B. A.O. was funded by a doctoral scholarship received from CIHR. J.L.M. was funded by a Master's award received from Defi Corporatif Canderel.

## Author contributions

A.O. contributed to the conceptualization, conducted the investigation, validated the experimental findings, wrote the original draft, and performed the formal analysis and visualization of experimental findings. C.B., and M.C.B. contributed to all in vivo investigation, analysis, and validation of experimental findings. J.L.M., X.J.L., and S.D.M. contributed to the investigation and validation of the senescence models, and with conceptualization, data analysis and helped edit and review the manuscript. C.B. contributed to the interpretation of the data and, furthermore reviewed and edited the manuscript. I.E.G. conceptualized, established and directed the execution of research goals, interpreted the data, reviewed and edited the manuscript.

## Competing interests

The authors declare no competing interests.
