## [Peer Review File · Nature Communications]

Reviewers' comments:

Reviewer #1 (Remarks to the Author):

The manuscript by Omer et al., is attempting to show that the RNA binding protein G3BP1 controls expression of SASP without impacting the induction of senescence and that it does so by impacting the ability of cGAS to bind to cytoplasmic chromatin fragments. This is an important area of investigation as senescence and its downstream SASP plays a role in various disease states including cancer, which is what is ultimately tested in this manuscript. While this is an important area there are several issues that should be addressed by the authors. Below I list these issues in no particular order of importance.

1. There are a couple of overstatements in the introduction that need to be addressed with primary literature citations (not references as was done) or they need to be removed or significantly "tuned down". For example:
 - a. It is stated on page 3 that "exposure to SASP is implicated as it is the main arbiter of many....". Only references are cited and whilst I think the authors are likely right I do not know that this has been shown. What has been shown is that elimination of senescent cells abrogates some age-related diseases; that doesn't mean it was the SASP that was responsible. Thus, unless the authors can offer primary literature to the contrary, I think this is overstated.
 - b. In the last paragraph they argue that they are using "in vivo models of senescence and cancer progression". I do not think that primary tumor growth of an injected tumorigenic line is tumor progression.
2. In several of the figures the examples shown do not appear to be representative of the graphed data (e.g. the SAHFs in the IF don't look like the graphs and the pictures of the tumors again do not look like the graph). To me representative photos should really show the mean of the graph or a range of pictures should be shown.
3. Figure 1 shows phospho-Rb and states that it remains phosphorylated but it does not. Please explain. Also, they are inducing senescence so why doesn't p16 go up and remain up? This has been explained/reported many times for this cell line under these conditions.
4. They state that fibroblasts depleted of G3BP1 (page 6) are "primed" for senescence. What does that mean (is this because they see lower levels of LMNB1 in these cells)? It looks like a subpopulation of these cells might already be senescent? If they wish to introduce a new term they should define what this means as I have never seen that description before and provide data.
5. Their induction of senescence based on the C12FDG (which can have lots of background), it would be important to show confirmatory data with the SABGAL stain, I realize SABGAL has its own background issues but having both would instill a bit more confidence. In any case their induction of senescence goes from ~55% in control cells to ~65% in siG3BP1 cells. This is not that impressive especially given they will soon argue that this results in large changes in SASP expression (if one has a culture of 55% senescent cells and one of 65%, what does that mean in regards to expression??? This doesn't seem to match).
6. Figure 1d, the presenescent siG3BP1 cells look to have just as many SAHFs as the senescent ones (just that their SAHFs are smaller...is this really representative or do the size of the SAHFs impact inclusion of a cell in the counts)? If so where is the rationale for having a size difference?
7. Figure 1H, what does it mean that the loss in LMNB1 is as large in regards to fold change in the presenescent to the senescent? Is this an argument for "primers cells"??
8. Figure S1C, the differences in BGal in the IMR90 is REALLY small. I see the "*" but is it biologically meaningful?

9. The authors try to claim that depletion of G3BP1 changes cGAS's ability to bind to cytoplasmic chromatin and this impacts SASP expression. There are several things that need to be addressed here:

- a. They should show that adding EGCG has no impact on siG3BP1 cells expression of SASP.
- b. They should double depleted cGAS and G3BP1 and see not additive effects on SASP expression.
- c. I don't really think their analysis of cGAS association with cytoplasmic fragments is that convincing. Can they do IPs and show a reduction? As to the staining, how do they know they aren't measuring mitochondrial DNA?

10. The heat maps in Figure 3 and 5, the numbers are impossible to read in the dark purple; if they wish to keep them considering changing the text to white or similar so the reader can actually read it.

11. The PCA graph, Figure 2a, why are the two replicates so different? They do not seem at all like replicates of a diploid cell line. This causes some concern about the factors labeled as SASP??

12. They state in the discussion that G3BP1 impacts the majority of the SASP but they only investigated inflammatory SASP factors so this seems like an overstatement that should be modified.

13. Figure 7 looks at the impact of senescent fibroblasts depleted of G3BP1 on tumor growth in vivo. They should show that the senescent fibroblasts are increasing growth over non senescent controls and then compare that to how G3BP1 depletion impacts growth...this should be in the in vitro work in figure 6 as well. Given the small changes this would help evaluate the magnitude of the impact.

14. I am not sure I have seen a two-way ANOVA used to assess growth differences in tumors over time. I would like to see the statistical justification for the use of the test especially given how small the growth differences are and how small the p values are.

15. THIS IS A MAJOR CONCERN: all of the work is based on one hairpin in two cell lines (although all the tumor data is only with one of these lines). They need to have either a second hairpin construct or they need to perform a rescue.

Reviewer #2 (Remarks to the Author):

In this manuscript entitled "The stress granule protein G3BP1 controls the senescence associated secretome and its impact on cancer progression", the authors report that G3BP1 plays an important role in cGAS-mediated induction of the senescent-associated secretory phenotype (SASP). Intriguingly, they find that the deletion of G3BP1 prevents SASP production without affecting the induction of senescence phenotype. The authors further employed in vitro and in vivo models to show G3BP1-depleted senescent cells impairs senescence-associated tumor growth. Overall this is a study reporting interesting findings, however, several points would require further evaluation through experiments, prior to any final recommendation.

MAJOR POINTS

1. The senescence model provided by this work is limited. The authors only employed primary human lung fibroblasts and induced senescence by exposing the cells to ionizing radiation (IR).

Additional models, or in vivo assays, if possible, are needed to further support the authors' conclusion.

2. The authors used one siRNA targeting G3BP1 to study the effect of G3BP1 on the SASP secretion or senescence. The effect of siRNA-mediated knockdown may be very limited, especially when the experiments last for 8 days to induce senescence after IR. Moreover, for the in vivo tumor growth assay, are the transfected siRNAs supposed to function throughout the 30 days during observation? A CRISPR-mediated G3BP1 deletion or MEFs from the G3bp1-KO murine embryos may provide more sufficient evidence to verify these findings. Also, the author should test the effect of G3BP1 in senescence induced by different stimuli.

3. Previous study (PMID: 28759028) reported that conditioned medium from oxidative stress-treated WT MEFs, but not cGAS KO MEFs, promoted a senescence response, and suggested that the role of cGAS in senescence mainly depends on secretion of soluble factors. In this work, the authors proposed that G3BP1 is required in the secretion of SASP mediated by cGAS, but not the induction of senescence. The authors should further evaluate the autocrine and paracrine effects of SASP from both WT and G3BP1-deficient cells.

4. The authors suggested that cGAS can promote senescence independently of its effect on the SASP. It's also possible that additional factor(s) that induces senescence may be induced in G3BP1-deficient cells. Deletion of both cGAS and G3BP1 in cells will help to address these questions.

5. Previous study (PMID: 29592859) showed that G3BP1 deficiency prevents the assembly of stress granules (SGs) and the arsenite-mediated effect of SGs on senescence. However, in this study, the author did not observe decreased senescence when G3BP1 was knocked down. Does ionizing radiation induce SGs formation? If it does, whether deletion of G3BP1 will affect the IR-induced SG formation? Also, do SGs affect the IR-related senescence and the activation of SASP secretion?

MINOR POINTS

1. According to the description of figure legends (Figures 1d-g, S1d-g), the immunofluorescence images seem to be inconsistent with the statistical graphs. A representative image should be shown. The authors need to check the similar issues throughout the paper.

2. Figure 4i is not a well-designed assay, many controls are lacking. Also, IRF3 is highly phosphorylated in "PRO" group, which are resting cells? The authors should confirm these data.

3. In Figure 5a, the "PRO" group also needs to be treated with EGCG to check whether the decreased expression of IRF3 and p65 induced by EGCG treatment is a senescent cell-limited phenomena or not.

Point-by-point rebuttal to reviewers' comments

We are grateful to the reviewers for the outstanding job evaluating our manuscripts and for the constructive comments and suggestions. We responded to all the comments made both experimentally and by amending the text.

In addition to the fully assembled manuscript, we are also providing a copy of the full manuscripts in which we highlight the changes made. These changes are highlighted in Yellow.

Below is a copy of the reviewer reports and our detailed answers. The reviewer comments are in normal font and are followed immediately by our responses.

Please find all our response to reviewer comments **bolded**.

Reviewers' comments:

Reviewer #1 (Remarks to the Author):

The manuscript by Omer et al., is attempting to show that the RNA binding protein G3BP1 controls expression of SASP without impacting the induction of senescence and that it does so by impacting the ability of cGAS to bind to cytoplasmic chromatin fragments. *This is an important area of investigation as senescence and its downstream SASP plays a role in various disease states including cancer, which is what is ultimately tested in this manuscript.* While this is an important area there are several issues that should be addressed by the authors. Below I list these issues in no particular order of importance.

1. There are a couple of overstatements in the introduction that need to be addressed with primary literature citations (not references as was done) or they need to be removed or significantly "tuned down". For example:

a. It is stated on page 3 that "exposure to SASP is implicated as it is the main arbiter of many...". Only references are cited and whilst I think the authors are likely right I do not know that this has been shown. What has been shown is that elimination of senescent cells abrogates some age-related diseases; that doesn't mean it was the SASP that was responsible. Thus, unless the authors can offer primary literature to the contrary, I think this is overstated.

We agree with the reviewer and have amended the text as suggested. Indeed, many secondary sources suggest that the SASP may be the arbiters of many of the deleterious effects of senescence. For this reason, we have tuned down the wording and modified the statement we originally made on page 3 as follows: "Despite the protective role that the senescence phenotype plays in an organism, the accumulation of senescent cells during aging have been associated with many cancers by enhancing neoplastic cell proliferation and metastasis"

b. In the last paragraph they argue that they are using "in vivo models of senescence and cancer progression". I do not think that primary tumor growth of an injected tumorigenic line is tumor progression.

We agree with the reviewer and we are grateful for this comment. We have changed all statements that read "cancer progression" or "tumor progression" to "cancer growth" or "tumor growth."

2. In several of the figures the examples shown do not appear to be representative of the graphed data (e.g. the SAHFs in the IF don't look like the graphs and the pictures of the tumors again do not look like the graph). To me representative photos should really show the mean of the graph or a range of pictures should be shown.

We thank the reviewer for this comment that we agree with. We have replaced a large number of images (Figures 1e, S1d-f, and 7e) to be more representative of our statistics.

3. Figure 1 shows phospho-Rb and states that it remains phosphorylated but it does not. Please explain. Also, they are inducing senescence so why doesn't p16 go up and remain up? This has been explained/reported many times for this cell line under these conditions.

We thank the reviewer for these comments. To address them, and to include the second siRNA against G3BP1, all these experiments were redone with quantifications (figures 1f, S3 and S4). Here we show that phosphorylation of RB is differentially affected depending on the cell line used or transduction method. However, in all cell line tested, the phosphorylation of RB was reduced independently of the means of senescence induction. Moreover, in our new blots and quantifications, we show, as is expected, that p16 protein levels increase upon induction of senescence and that these levels are maintained throughout the process. We describe these effects on pages 4 and 5 in our revised manuscript.

4. They state that fibroblasts depleted of G3BP1 (page 6) are "primed" for senescence. What does that mean (is this because they see lower levels of LMNB1 in these cells)? It looks like a subpopulation of these cells might already be senescent? If they wish to introduce a new term they should define what this means as I have never seen that description before and provide data.

We thank the reviewer for raising this point and we totally agree with the comment. Since the depletion of G3BP1 does not affect cell commitment to senescence, our study focuses on the impact of G3BP1 on several senescence associated phenotypes such as SASP secretion. We apologize for any confusion and have modified our text accordingly, as can be seen on pages 5 and 6 of our revised manuscript, so that we no longer make reference to the cells being "primed."

5. Their induction of senescence based on the C12FDG (which can have lots of background), it would be important to show confirmatory data with the SABGAL stain, I realize SABGAL has its own background issues but having both would instill a bit more confidence. In any case their induction of senescence goes from ~55% in control cells to ~65% in siG3BP1 cells. This is not that impressive especially given they will soon argue that this results in large changes in SASP expression (if one has a culture of 55% senescent cells and one of 65%, what does that mean in regards to expression??? This doesn't seem to match).

We thank the reviewer for this comment. We have added, in our revised manuscript, new data assessing SA-beta-galactosidase staining in Figures 1c and S2c. These new results were similar to those achieved by performing IF experiments using C12FDG staining. As mentioned in our response to point 4 (see above), we agree with the reviewer that the impact of G3BP1 depletion on cell commitment to senescence is not impressive (10-15%). Therefore, our data clearly show that even though G3BP1 depleted cells are able to senesce, they lose their ability to secrete SASP. Our data also show that the loss of SASP on its own is enough to prevent these senescent cells from promoting

paracrine effects of senescence such as promotion of cancer growth and paracrine-induction of senescence. We amended the text throughout our manuscript to clarify this point.

6. Figure 1d, the pre-senescent siG3BP1 cells look to have just as many SAHFs as the senescent ones (just that their SAHFs are smaller...is this really representative or do the size of the SAHFs impact inclusion of a cell in the counts)? If so where is the rationale for having a size difference?

We thank the reviewer for this comment. The interpretation of our data are based on those presented by Zhang *et al* (Developmental Cell 2005) where they defined SAHF as "DAPI foci" and size was not considered. We have further added, in this revised manuscript, data demonstrating staining by H3K9me3 (a marker of heterochromatin). "SAHF positive" cells, in our study, are characterized as those demonstrating the colocalization of foci formed by both stains (as seen in Figure 1e of the revised manuscript).

7. Figure 1H, what does it mean that the loss in LMNB1 is as large in regards to fold change in the presenescent to the senescent? Is this an argument for "primers cells"??

We thank the reviewer for this comment and have addressed this concern as stated above.

8. Figure S1C, the differences in β Gal in the IMR90 is REALLY small. I see the "*" but is it biologically meaningful?

We thank the reviewer for this comment and have addressed this concern as stated above.

9. The authors try to claim that depletion of G3BP1 changes cGAS's ability to bind to cytoplasmic chromatin and this impacts SASP expression. There are several things that need to be addressed here:
a. They should show that adding EGCG has no impact on siG3BP1 cells expression of SASP.

We thank the reviewer for suggesting this important experiment and have shown (Figure 5d of revised manuscript) that addition of EGCG to G3BP1 depleted cells does not further diminish the SASP, demonstrating that EGCG acts through G3BP1.

b. They should double depleted cGAS and G3BP1 and see not additive effects on SASP expression.

We thank the reviewer for this comment, and we agree that the suggested experiment is an important control. We depleted both cGAS and G3BP1 (see Figure S10e of revised manuscript) and show that the simultaneous loss of both factors has no additive effect on SASP expression.

c. I don't really think their analysis of cGAS association with cytoplasmic fragments is that convincing. Can they do IPs and show a reduction? As to the staining, how do they know they aren't measuring mitochondrial DNA?

These are valid comments that we address as follow: While immunoprecipitating cGAS and assessing reduction in association with CCFs is a good suggestion, in our study, we followed best practice for quantification of cGAS-CCF colocalization (Dou Z, *Nature* 2017; Gluck S, *Nature Cell Bio* 2017; Liu ZS, *Nature Imm* 2018; Hu S, *Science Signaling* 2019; Yang H, *PNAS*, 2017; Jiang H, *EMBO* 2019).

As for the staining, we are confident that our analysis likely identifies cGAS colocalized with CCFs. Indeed, these nuclear derived CCF are described as large co-localized foci containing DAPI, cGAS and importantly gamma-H2AX. Since, as far as we know, mitochondrial DNA do not contain histones and therefore do not demonstrate co-staining with gamma-H2AX, we believe that this association occurs with gDNA.

10. The heat maps in Figure 3 and 5, the numbers are impossible to read in the dark purple; if they wish to keep them considering changing the text to white or similar so the reader can actually read it.

We thank the reviewer for this comment. We agree with the reviewer and, as such, our multiplex heatmaps have been amended through increasing the transparency of the colors to clearly show the text (Figures 3b and 5c of revised manuscript).

11. The PCA graph, Figure 2a, why are the two replicates so different? They do not seem at all like replicates of a diploid cell line. This causes some concern about the factors labeled as SASP??

We thank the reviewer for this comment. We have adjusted the y-axis to the figure (Figure 2a) to more clearly demonstrate the grouping. However, to clarify, the PCA analysis was not used to discern which factors were labeled as SASP, but to ensure that we saw global differences between our siCTL and siG3BP1 treatments, which we clearly demonstrate. To be precise, we used differential expression analysis to discern the impact on SASP independently from our PCA analysis, which we clarified in our revised manuscript (page 6). Furthermore, we validated our findings by RT-qPCR using multiple siRNAs in different cells lines and different means of senescence induction, as suggested by the reviewer, and using the G3BP1 inhibitor EGCG.

12. They state in the discussion that G3BP1 impacts the majority of the SASP but they only investigated inflammatory SASP factors so this seems like an overstatement that should be modified.

We agree with the reviewer and have clarified the revised manuscript whereby we refer to these factors as inflammatory SASP as seen in pages 6 and 7 of the revised manuscript.

13. Figure 7 looks at the impact of senescent fibroblasts depleted of G3BP1 on tumor growth in vivo. They should show that the senescent fibroblasts are increasing growth over non senescent controls and then compare that to how G3BP1 depletion impacts growth...this should be in the in vitro work in figure 6 as well. Given the small changes this would help evaluate the magnitude of the impact.

[REDACTED]

[REDACTED]

[REDACTED]

14. I am not sure I have seen a two-way ANOVA used to assess growth differences in tumors over time. I would like to see the statistical justification for the use of the test especially given how small the growth differences are and how small the p values are.

We agree with the reviewer. The growth curves, tumor volumes and integrated density curves included in Fig. 6, S11 and 7 are now only represented by R-squared values and means, and we have run one-way anovas on all end-point measurements (Figure 6b, S11, and 7c-e).

15. THIS IS A MAJOR CONCERN: all of the work is based on one hairpin in two cell lines (although all the tumor data is only with one of these lines). They need to have either a second hairpin construct or they need to perform a rescue.

We agree with the reviewer and have reproduced the majority of our key findings with a second siRNA targeting G3BP1 (Figure 1, 3a and c, 4a and b, S2, S5, S7 and S9).

Reviewer #2 (Remarks to the Author):

In this manuscript entitled “The stress granule protein G3BP1 controls the senescence associated secretome and its impact on cancer progression”, the authors report that G3BP1 plays an important role in cGAS-mediated induction of the senescent-associated secretory phenotype (SASP). Intriguingly, they find that the deletion of G3BP1 prevents SASP production without affecting the induction of senescence phenotype. The authors further employed in vitro and in vivo models to show G3BP1-depleted senescent cells impairs senescence-associated tumor growth. Overall this is a study reporting interesting findings, however, several points would require further evaluation through experiments, prior to any final recommendation.

MAJOR POINTS

1. The senescence model provided by this work is limited. The authors only employed primary human lung fibroblasts and induced senescence by exposing the cells to ionizing radiation (IR). Additional models, or in vivo assays, if possible, are needed to further support the authors’ conclusion.

We thank the reviewer for this comment. We agree with the reviewer and, as such, have included new data in our revised manuscript demonstrating that oncogene-induced senescence recapitulates our observations using ionizing radiation-induced senescence (Figure S2, S5, S7 and S9).

2. The authors used one siRNA targeting G3BP1 to study the effect of G3BP1 on the SASP secretion or senescence. The effect of siRNA-mediated knockdown may be very limited, especially when the experiments last for 8 days to induce senescence after IR. Moreover, for the in vivo tumor growth assay, are the transfected siRNAs supposed to function throughout the 30 days during observation? A CRISPR-mediated G3BP1 deletion or MEFs from the G3bp1-KO murine embryos may provide more sufficient evidence to verify these findings. Also, the author should test the effect of G3BP1 in senescence induced by different stimuli.

We thank the reviewer for this comment. We have added data demonstrating that knockdown of G3BP1 lasts throughout the 30 days used in our in vivo experiments (Figure S13a).

3. Previous study (PMID: 28759028) reported that conditioned medium from oxidative stress-treated WT MEFs, but not cGAS KO MEFs, promoted a senescence response, and suggested that the role of cGAS in senescence mainly depends on secretion of soluble factors. In this work, the authors proposed that G3BP1 is required in the secretion of SASP mediated by cGAS, but not the induction of senescence. The authors should further evaluate the autocrine and paracrine effects of SASP from both WT and G3BP1-deficient cells.

We thank the reviewer for this comment. We agree with the reviewer and have, as suggested, performed experiments demonstrating that depletion of G3BP1 impairs paracrine signalling, thereby abrogating the senescence promoting effects in proliferative WI-38 cells (Figure S12).

4. The authors suggested that cGAS can promote senescence independently of its effect on the SASP. It’s also possible that additional factor(s) that induces senescence may be induced in G3BP1-deficient cells. Deletion of both cGAS and G3BP1 in cells will help to address these questions.

We thank the reviewer for this suggestion. We have included in the revised manuscript new data demonstrating that the double-knockdown of cGAS and G3BP1 does not abrogate cellular senescence (Figure S10 of revised manuscript).

5. Previous study (PMID: 29592859) showed that G3BP1 deficiency prevents the assembly of stress granules (SGs) and the arsenite-mediated effect of SGs on senescence. However, in this study, the author did not observe decreased senescence when G3BP1 was knocked down. Does ionizing radiation induce SGs formation? If it does, whether deletion of G3BP1 will affect the IR-induced SG formation? Also, do SGs affect the IR-related senescence and the activation of SASP secretion?

We thank the reviewer for this comment. We have included new data demonstrating that ionizing radiation, unlike chronic oxidative stress, does not induce stress granule (SG) formation. As such, our data demonstrates that the function of G3BP1 in IR-related senescence occurs independently of SGs (Figure S5).

MINOR POINTS

1. According to the description of figure legends (Figures 1d-g, S1d-g), the immunofluorescence images seem to be inconsistent with the statistical graphs. A representative image should be shown. The authors need to check the similar issues throughout the paper.

We thank the reviewer for this comment. As per our response to reviewer 1, we have replaced a large number of images (Figure 1e, S1d-f, and 7e) to be more representative of our statistics.

2. Figure 4i is not a well-designed assay, many controls are lacking. Also, IRF3 is highly phosphorylated in “PRO” group, which are resting cells? The authors should confirm these data.

We thank the reviewer for this comment. As controls, we included the cGAMP treated proliferative controls (with and without siG3BP1) and the senescent WI-38 treated with siCTL and assessed the status of IRF3, NF- κ B and STAT3 pathways in these cells. As expected, although these pathways were activated in these senescent cells, they were downregulated when G3BP1 was depleted.

We also thank the reviewer for the comment on IRF-3 phosphorylation. In our hands, we observed that IRF-3 is always activated in proliferative WI-38 treated with siRNA. The explanation for this effect is still unknown. However, our data clearly show that IRF-3 phosphorylation depends on G3BP1 during senescence and that addition of the cGAS product cGAMP, re-established, in part, this phosphorylation.

3. In Figure 5a, the “PRO” group also needs to be treated with EGCG to check whether the decreased expression of IRF3 and p65 induced by EGCG treatment is a senescent cell-limited phenomena or not.

We thank the reviewer for this comment and have added proliferative cells treated with EGCG to our analysis (Figure 5a).

REVIEWER COMMENTS

Reviewer #2 (Remarks to the Author):

The authors have improved the manuscript during last revision. However, there are still few concerns that need to be further addressed prior to a recommendation for publication in Nature Communication.

1. The data regarding the effect of cGAS-knockdown on both SASP induction and cellular senescence supported the claim that G3BP1 depletion impairs SASP induction without inhibiting senescence in this specific cell type. However, according to previous studies (PMID: 28759028, 28533362), cGAS deletion abrogates the senescence in MEF cells. Knockdown of cGAS and G3BP1 in MEF cells seems to be necessary to determine whether the effect of cGAS and G3BP1 on cellular senescence are cell type-dependent.
2. As shown in Fig 4h "PRO" group, cGAMP almost had no effect on the activation of cGAS-downstream signaling, as indicated by phospho-IRF3. This result suggested that the cells were unresponsive to cGAMP stimulation. How could supplementation of cGAMP rescued the dampened SASP induction in G3BP1-deleted senescent cells (Fig. 4i)?
3. The corresponding data are missing according to the description of Figure 4i (page 8; second to last line).

Point-by point rebuttal to reviewer comments

We are appreciative to the reviewers for their comprehensive evaluation of our manuscripts and for their constructive comments and suggestions. We responded to all the comments made by the reviewers both experimentally and by amending the text.

In addition to the fully assembled manuscript, we are also providing a copy of the full manuscripts in which we highlight the changes made. These changes are highlighted in Yellow.

Below is a copy of the reviewer comments and our point-by-point answers. The reviewer comments are in normal font and are followed immediately by our responses.

Please find all our response to reviewer comments bolded.

Reviewer #2 (Remarks to the Author):

The authors have improved the manuscript during last revision. However, there are still few concerns that need to be further addressed prior to a recommendation for publication in Nature Communication.

1. The data regarding the effect of cGAS-knockdown on both SASP induction and cellular senescence supported the claim that G3BP1 depletion impairs SASP induction without inhibiting senescence in this specific cell type. However, according to previous studies (PMID: 28759028, 28533362), cGAS deletion abrogates the senescence in MEF cells. Knockdown of cGAS and G3BP1 in MEF cells seems to be necessary to determine whether the effect of cGAS and G3BP1 on cellular senescence are cell type dependent.

We agree with the reviewer that these effects are likely cell type dependent. While we believe that comparing these cells' ability to enter senescence is important, our study intends to assess the effects of G3BP1 on SASP, and not senescence induction. Additionally, as shown in the studies mentioned by the reviewer (PMID: 28759028, 28533362), MEFs are able to spontaneously immortalize and escape senescence unlike human cells, which may lend to the observed effects within those studies. Indeed, in PMID: 28759028, the authors looked at this effect in WI-38 cells and did see a slight but non-significant decrease in senescence with loss of cGAS (note Figure 2F and Supplemental Figure 4B from PMID: 28759028). Consistent with this observation, we did not observe any change in senescence induction in our WI-38 cells depleted of cGAS (Supplementary Fig. 10). Therefore, we agree that the effect of cGAS depletion on senescence induction are cell-type/species specific.

2. As shown in Fig 4h "PRO" group, cGAMP almost had no effect on the activation of cGAS-downstream signaling, as indicated by phospho-IRF3. This result suggested that the cells were unresponsive to cGAMP stimulation. How could supplementation of cGAMP rescued the dampened SASP induction in G3BP1-deleted senescent cells (Fig. 4i)?

We thank the reviewer for this comment. We agree that the PRO group appears unresponsive to cGAMP. We believe that this is simply due to the incomplete knockdown of G3BP1 in proliferative cells. In all of our experiments, the efficiency of G3BP1 knockdown *in proliferative cells* is lower than what is seen in senescent cells. Indeed, as seen in a number of our blots (Figures 1F and 4H, Supp Figs 1A, 3A and 4A), the knockdown of G3BP1 is more effective during senescence than during

proliferation. In addition, in our experiments, the proliferative cells exhibited high baseline IRF3 phosphorylation (which is probably at saturation level) that was not enhanced by addition of exogenous cGAMP. However, in senescent cells where the knockdown of G3BP1 is nearly 100%, we clearly see a reduction in IRF3 phosphorylation, which was reverted by addition of cGAMP (Fig 4i).

3. The corresponding data are missing according to the description of Figure 4i (page 8; second to last line).

We thank the reviewer for this comment and have amended the text.

REVIEWERS' COMMENTS:

Reviewer #2 (Remarks to the Author):

The reviewer has no further suggestions.